# Membrane curvature regulates the spatial distribution of bulky glycoproteins

Chih-Hao Lu[1], Kayvon Pedram[1,4], Ching-Ting Tsai[1], Taylor Jones IV[1], Xiao Li[1,5], Melissa L. Nakamoto[1], Carolyn R. Bertozzi ● [1,2,3] & Bianxiao Cui ● [1✉]

The glycocalyx is a shell of heavily glycosylated proteins and lipids distributed on the cell surface of nearly all cell types. Recently, it has been found that bulky transmembrane glycoproteins such as MUC1 can modulate membrane shape by inducing membrane protrusions. In this work, we examine the reciprocal relationship of how membrane shape affects MUC1's spatial distribution on the cell membrane and its biological significance. By employing nanopatterned surfaces and membrane-sculpting proteins to manipulate membrane curvature, we show that MUC1 avoids positively-curved membranes (membrane invaginations) and accumulates on negatively-curved membranes (membrane protrusions). MUC1's curvature sensitivity is dependent on the length and the extent of glycosylation of its ectodomain, with large and highly glycosylated forms preferentially staying out of positive curvature. Interestingly, MUC1's avoidance of positive membrane curvature enables it to escape from endocytosis and being removed from the cell membrane. These findings also suggest that the truncation of MUC1's ectodomain, often observed in breast and ovarian cancers, may enhance its endocytosis and potentiate its intracellular accumulation and signaling.

[1] Department of Chemistry, Stanford University, Stanford, CA 94305, USA. [2] Stanford ChEM-H, Stanford University, Stanford, CA 94305, USA. [3] Howard Hughes Medical Institute, Stanford University, Stanford, CA 94305, USA. [4]Present address: Janelia Research Campus, Howard Hughes Medical Institute, Ashburn, VA 20147, USA. [5]Present address: School of Mechanical Engineering, Xi'an Jiaotong University, Xi'an, China. ✉email: bcui@stanford.edu

Cell membranes are dynamic and readily bent by membrane-sculpting proteins and cytoskeleton forces. Many vital cellular processes require precise manipulation of the membrane shape or membrane curvature[1,2]. For example, clathrin-mediated endocytosis, an essential cellular process conserved from yeast to humans, involves gradual bending of the plasma membrane inward assisted by proteins that contain N-BAR (Bin/Amphiphysin/Rvs), F-BAR, or ENTH domains that induce positive membrane curvature[3–6]. Intestinal cells organize their membranes into microvilli to increase surface area for nutrient absorption[7,8], and cancer cells overexpress inverse BAR (I-BAR) proteins to create protrusive filopodia for migration and cell invasion[9].

Nearly all mammalian cells are coated with a layer of heavily glycosylated compartments, collectively termed the glycocalyx. The glycocalyx has been documented to play active roles in both biophysical and immunological pathways in the progression of diseases ranging from respiratory viral infections to carcinomas[10]. Transmembrane mucin, MUC1 (Mucin 1, also known as episialin), is one of the major components of the mammalian glycocalyx, and serves as a protective barrier between the cell and its surroundings. Its large N-terminal extracellular domain with heavily glycosylated branches renders a bottle-brush structure that can extend hundreds of nanometers from the cell surface. The short cytoplasmic domain of MUC1 can be processed and transported to the nucleus to regulate a number of transcription factors and signaling pathways[11]. MUC1's unique physical property not only imparts physical barriers to provide protective functions for the cell membrane[12–17] but also conveys biochemical signals for modulating cell-cell, cell-matrix, and ligand-receptor interactions[12,13,18,19]. Interestingly, MUC1 is overexpressed but underglycosylated in many cancer types and is believed to activate multiple signaling pathways in cancer cells[13,20,21]. It is believed that a reduction in glycosylation level destabilizes the extracellular domain, which facilitates the translocation of the C-tail into the nucleus[20], but the underlying molecular mechanism is yet to be fully elucidated.

Notably, large glycosylated proteins appear to affect the shape of the plasma membrane. MUC1 and other large glycosylated proteins have been shown to accumulate at high densities on protrusive membrane structures such as filopodia[22,23], microvilli[14,24–26], and cilia[27]. Paszek and co-workers found that a bulky glycocalyx facilitates integrin clustering by funneling active integrins into adhesions and applying mechanical tension to them, thus mediating integrin-dependent cell adhesion and survival[28]. Interestingly, a follow-up investigation demonstrated that overexpression of bulky glycocalyx polymers is sufficient to induce membrane protrusions in a density-dependent and length-dependent manner[29]. By deleting the intracellular domain and using synthetic polymer backbones, the authors further demonstrate that the membrane-bending effect is not due to intracellular protein signaling, but due to the entropic force induced by protein-protein crowding[29–31].

Although it is now established that overexpression of bulky glycocalyx proteins can induce membrane protrusions, the reciprocal relationship, i.e. whether the shape of the plasma membrane such as inward invaginations (positive curvature) and outward protrusions (negative curvature), affects the spatial distribution of the glycocalyx and its intracellular signaling, is yet to be carefully investigated. Due to the physical attachment of the glycocalyx to the cell membrane, the extracellular bulky domain of glycopolymers such as MUC1 may be able to sense membrane curvature and redistribute on the plasma membrane. Recent studies show that, due to conformational entropy, intrinsically disordered regions (IDRs) are able to sense membrane curvature when artificially tethered to the membrane and can amplify the curvature sensitivity of F-BAR domains[32–34]. Therefore, MUC1 may sense membrane curvatures via a similar mechanism.

In this work, we systematically examine how membrane curvature affects MUC1's spatial distribution. To manipulate membrane curvature, we take advantage of two independent approaches. The first approach uses vertical nanotopography to imprint the plasma membrane with positive or negative curvature. The second approach uses the overexpression of membrane-sculpting proteins for biologically-induced positively- and negatively-curved membranes. Our results show that large and highly glycosylated MUC1 variants avoid positively-curved membranes. The avoidance of positive membrane curvature leads to reduced MUC1 internalization, which likely contributes to its long lifetime on the cell membrane.

## Results

**MUC1 avoids positively-curved membranes induced by vertical nanobars and nanopillars.** To induce well-defined plasma membrane curvature, we engineered SiO$_2$ nanopillar arrays using photolithography followed by wet etching to shrink the feature size (Fig. 1A, 200 nm in diameter, 1 μm in height, and 2.5 μm in spacing)[35,36]. Previous electron microscopy studies by us and others[37–39] show that when cells are cultured on substrates with nanopillar arrays of these dimensions, the plasma membrane wraps around nanopillars to create local membrane curvature (Fig. 1B). Nanopillar-induced membrane curvature is membrane invagination, which is defined as positive curvature. The curvature value is determined by the diameter of the nanopillars. We also engineered vertical nanobars that induce two different local membrane curvature profiles—highly positive curvature at the two ends and relatively flat membranes along the sides (200 nm in width, 2 μm in length, 1 μm in height, and 5 μm in spacing, Fig. 1C, D). For most of our imaging studies, we focus near the middle height of the nanostructures.

To probe the contribution of MUC1's glycosylated ectodomain to its observed curvature sensitivity, we employed a previously reported toolkit of MUC1 constructs[40]. The extracellular domain of a native MUC1 construct consists of 42 tandem repeats (TR) of a heavily glycosylated peptide PDTRPAPGSTAPPAHGVTSA. We utilized the MUC1_42TR-GFP construct[40] that is composed of 42 native TRs, a proximal oxidizing environment-optimized variant of green fluorescent protein (mOxGFP, simply denoted as "GFP" below for convenience) for imaging, and a native MUC1 transmembrane domain (Fig. 1E). The cytoplasmic tails for all MUC1 constructs used in this work are deleted in order to avoid MUC1-dependent intracellular signaling (MUC1-ΔCT, simply denoted as 'MUC1' below for convenience)[40]. MUC1 constructs with shorter glycocalyx ectodomains, including MUC1_21TR-GFP and MUC1_10TR-GFP, were used to study the effects of ectodomain length. Another construct MUC1_0TR-GFP that lacks any tandem repeats and thus cannot be glycosylated was used as a negative control. Flow cytometric analysis shows similar expression levels of these MUC1 constructs on the surface of U2OS cells (Supplementary Fig. 1).

We co-transfected U2OS cells with a MUC1 construct and a membrane marker mCherry-CAAX[41]. Clathrin adapter protein AP2 and F-actin that have been previously shown to accumulate at positively-curved membranes around nanopillars[5,35] were used as positive controls. AP2 and F-actin were (immuno)stained with specific antibodies and phalloidin, respectively. When transfected cells were cultured on nanobar substrates, mCherry-CAAX signals show that cell membranes wrap around these nanobars with relatively uniform distribution along the length of nanobars (Fig. 1F, G). On the same nanobars, MUC1_42TR-GFP appears to distribute more on the flat side walls (zoom-in image in

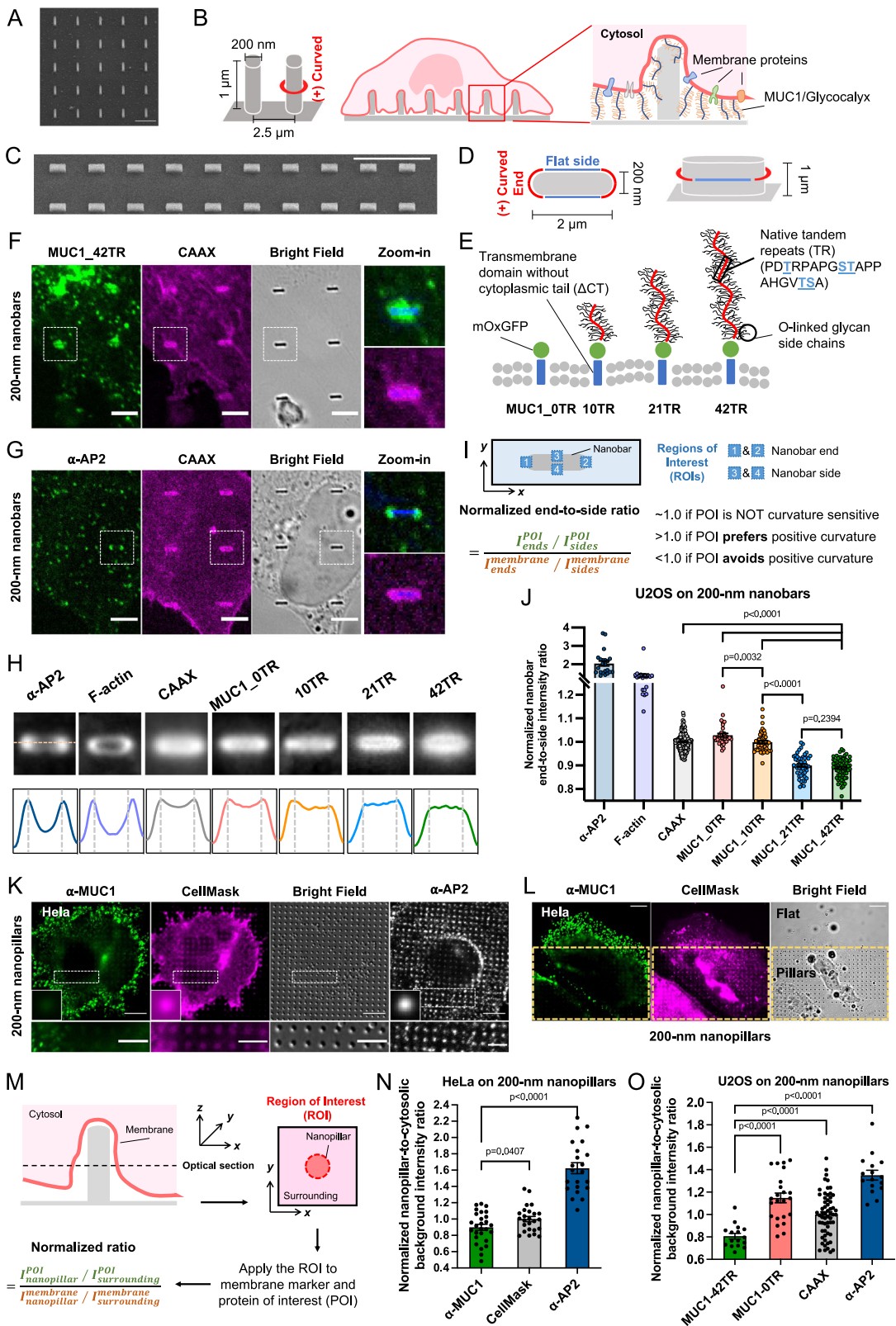

Fig. 1F), while AP2 strongly favors the two ends of nanobars with highly curved membranes (zoom-in image in Fig. 1G). There are large variations among protein responses to individual nanobars, therefore, we averaged hundreds to thousands of nanobars to obtain an average image for each protein (Details of data processing shown in Supplementary Fig. 2). The averaged fluorescence images and intensity profiles show that AP2 and F-actin, two proteins previously demonstrated to prefer positive curvature, strongly accumulate at the ends of nanobars (Fig. 1H). Compared with the membrane marker CAAX, MUC1_42TR and MUC1_21TR show reduced signals at the nanobar ends (Fig. 1F–H and Supplementary Fig. 3). We quantified the nanobar end-to-side ratio for each protein in individual cells and then normalized the ratios to the average value of CAAX

**Fig. 1 MUC1 avoids positively-curved membranes induced by vertical nanobars and nanopillars. A** A scanning electron microscopy (SEM) image (tilted at 45°) of the 200-nm-diameter nanopillar arrays. **B** Schematic illustration of positive membrane curvature induced by nanopillars. **C** An SEM image (tilted at 45°) of 200-nm-wide nanobar arrays. **D** Schematic illustration of a nanobar that induces positive membrane curvature at two ends and flat membranes along the side walls. **E** Schematic illustration of MUC1-△CT-mOxGFP constructs with varying numbers of tandem repeats (TRs). **F, G** Confocal images of (**F**) U2OS cells co-transfected with MUC1_42TR-GFP and mCherry-CAAX or (**G**) mCherry-CAAX-transfected U2OS cells stained with anti-AP2 on 200-nm nanobar arrays. Bright-field images of nanobars in the zoom-in subsets were converted into blue colors for visualization. **H** Averaged fluorescence images and intensity profiles show the spatial distributions of MUC1 variants of varying lengths, mCherry-CAAX, AP2, and F-actin on 200-nm nanobar arrays. **I** Illustrations of the quantification process for the normalized nanobar end-to-side ratio. **J** Quantification of nanobar end-to-side ratios of AP2, F-actin, and MUC1-GFP of different lengths, normalized to the mean ratio of CAAX. **K** Confocal images of Hela cells cultured on the 200-nm nanopillar arrays. Membranes were visualized via CellMask staining and MUC1 was immunostained with anti-MUC1. In a separate experiment, cells were stained with anti-AP2 that serves as a control. The insets are the averaged images of proteins at nanopillars. **L** A Hela cell cultured at the nanopillar-flat boundary. **M** Illustrations of the quantification process for the normalized nanopillar-to-surrounding ratio. **N** Quantification of nanopillar-to-surrounding ratios of α-MUC1, CellMask, and AP2 signals in Hela cells on the 200-nm nanopillar arrays. **O** Quantification of nanopillar-to-surrounding ratios for MUC1_42TR-GFP, MUC1_0TR-GFP, mCherry-CAAX, and AP2 signals in U2OS cells. See Supplementary Tables 1 and 2 for the detailed statistics. Scale bars = 10 μm for **C, K, L**; Scale bars = 5 μm for **F, G**, and zoom-in images in **K**; Scale bars = 2 μm for **A**. Welch's *t*-tests (unpaired, two-tailed, not assuming equal variance) are applied for all statistical analyses in this figure. Error bars represent SEM.

(Fig. 1I, Supplementary Fig. 2, and Supplementary Table 1). These quantifications confirm that MUC1-42TR and MUC1_21TR, both with large ectodomains, have reduced presence at positive membrane curvature at the nanobar ends (Fig. 1J). The shorter isoforms, MUC1-10TR and MUC1_0TR, behave similarly to the membrane marker CAAX (Fig. 1J and Supplementary Fig. 3).

We further probed the distribution of endogenous MUC1 on nanopillars by immunostaining HeLa cells that express a high level of MUC1. Hela cells were stained with anti-MUC1 as well as CellMask dye to label the plasma membrane. CellMask can also stain intracellular membranes at a low intensity, thus giving an overall higher background than CAAX. Fluorescence images show that AP2 and F-actin preferentially accumulate at nanopillar locations (Fig. 1K and Supplementary Fig. 4A, B), while anti-MUC1 shows less signal at nanopillar locations than the membrane marker CellMask. Quantification of the nanopillar-to-surrounding ratio, normalized to the ratio of CellMask signal (Fig. 1M and Supplementary Fig. 2), shows that the ratio for endogenous MUC1 is ~0.9, while the ratios for AP2 and F-actin are ~1.6 and ~1.25, respectively (Fig. 1N, Supplementary Table 2A, and Supplementary Fig. 4C). These quantifications confirm a reduced presence of endogenous MUC1 at positively-curved membranes surrounding nanopillars. Interestingly, when cells are at the nanopillar/flat boundary, MUC1 appeared to prefer the flat region rather than the nanopillar region (Fig. 1L). Previous studies show that MUC1 in Hela cells largely localizes on filopodia protrusions[22]. We speculate that nanopillars create positive membrane curvatures, which makes it unfavorable for filopodia formation that requires the generation of negative membrane curvatures.

We also cultured transfected U2OS cells on nanopillars and quantified the nanopillar-to-surrounding ratios. By normalizing the protein ratio against the membrane signal (CAAX), we found that MUC1_42TR-GFP displays a nanopillar-to-surrounding ratio of about 0.8, indicating that MUC1_42TR-GFP accumulates less at nanopillars than the membrane marker, in contrast to anti-AP2 (~1.35) and F-actin (~1.2) which localize more to nanopillars (Fig. 1O, Supplementary Table 2B, and Supplementary Fig. 5). MUC1_0TR-GFP displays a nanopillar-to-surrounding ratio ~1.15 (Fig. 1O and Supplementary Fig. 5), agreeing with the hypothesis that the diminished presence of MUC1_42TR-GFP on positively-curved membranes is due to its bulky ectodomain.

**MUC1 prefers negatively-curved over positively-curved membranes on the same nanostructures.** To further determine the

curvature preference of MUC1, we fabricated vertical nanostructures that can induce both positive and negative curvatures. For this purpose, we designed and fabricated gradient NanoX arrays which are able to induce positive curvature at the arm ends and negative curvature at the inner faces (Fig. 2A). The gradient nanoX arrays were designed with inner angles (θ) ranging from 30° to 90° nm with a 15° increment (350 nm in arm width and 5 μm in arm length, 2 μm in height, and 10 μm in spacing, Fig. 2B). A nanoX with a 30° inner angle also possesses a complementary angle 150°.

Phalloidin staining shows that F-actin preferentially accumulates at the ends of nanoX arms, where the plasma membrane is positively-curved (Fig. 2C, D). In contrast, on the same nanoXs, MUC1_42TR-GFP accumulates more at the inner faces with negative curvature compared to the CAAX membrane control. Averaging over many nanoX structures, F-actin, mCherry-CAAX, MUC1_0TR, and MUC_42TR show different spatial distributions (Fig. 2E and Supplementary Figs. 6, 7). To quantify curvature preference, we evaluated the ratio of fluorescence intensities at the arm ends (positive curvature) or the inner faces (negative curvature) to the side walls (flat) of nanoX (Supplementary Fig. 2 and Supplementary Table 3). After normalizing the protein ratios to the membrane ratio, MUC1-42TR shows a preference for the negative curvature with higher inner-to-side ratios (Fig. 2F), in contrast to F-actin that prefers positive curvature with higher end-to-side ratios (Fig. 2G), and to MUC1_0TR-GFP that shows similar ratios to CAAX (Supplementary. Fig. 8). The plasma membrane adheres to the ends of nanoX, but likely not tightly to the inner surfaces of nanoXs. This can be seen from mCherry-CAAX signals (Fig. 2E and Supplementary Figs. 6B, 7), which shows a much higher intensity at the four ends than at the inner faces. Although negative curvature is induced at the inner faces of nanoXs, the curvature value is not defined by the angle, which explains why there is no apparent difference between the two complementary inner faces in nanoXs (Fig. 2F, G).

MUC1's preference for negatively-curved membranes was independently demonstrated by a dense nanopillar array (Fig. 2H, 1 μm in spacing, 1 μm in height, and 500 nm in diameter), where the MUC1 immunostaining in HeLa cells was shown to be spatially anti-correlated with nanopillars (Fig. 2I). This phenomenon was not observed when HeLa cells were cultured on sparse nanopillar arrays with 2.5-μm or 5-μm-spacing (Supplementary Fig. 9). Using transmission electron microscopy, we have previously shown that cells adhere to the bottom surface and wrap around thin and sparse nanopillars (200 nm in diameter and 2.5 μm in spacing), but stay at the top of dense and large nanopillar arrays (500 nm in diameter and 1 μm in spacing, same

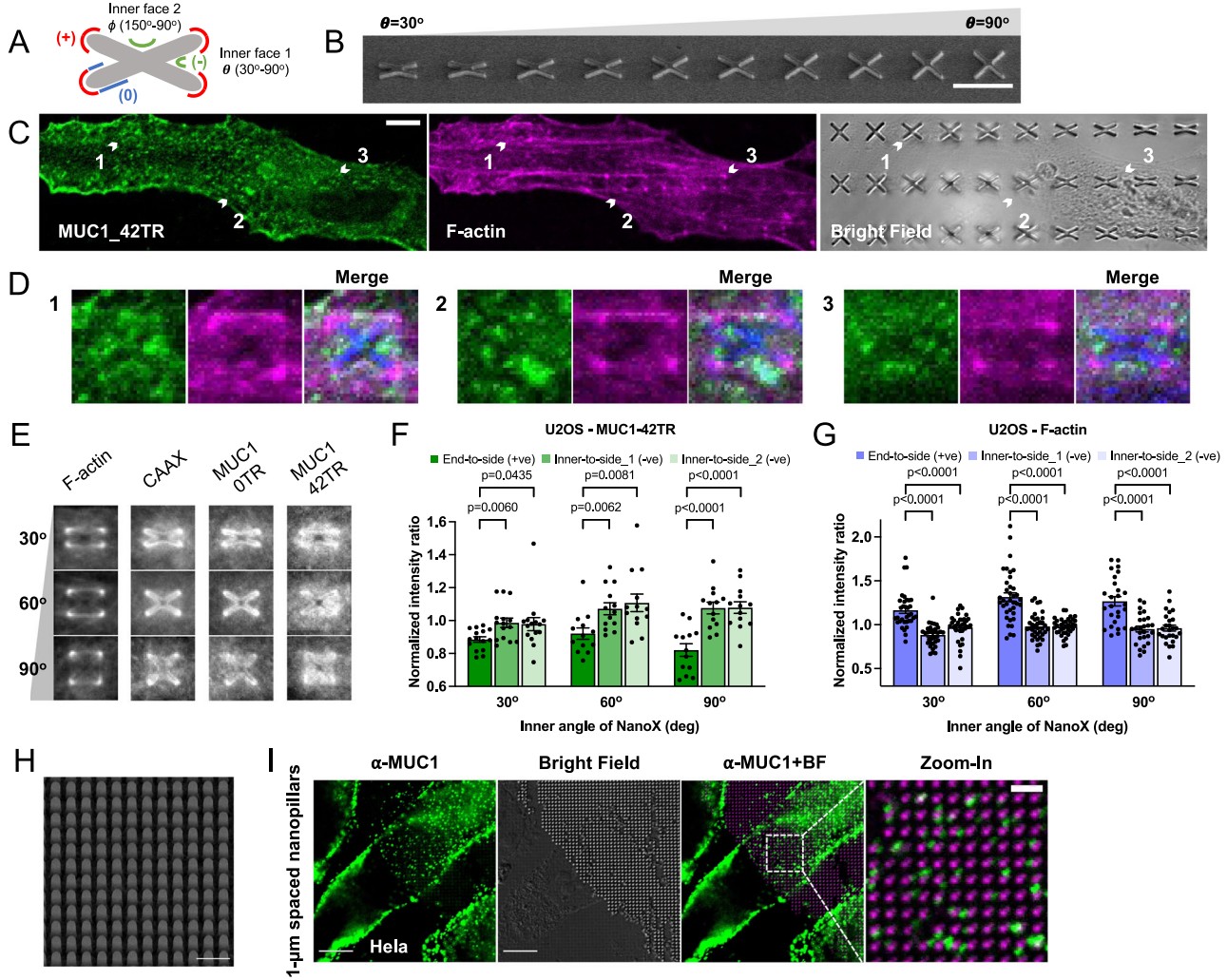

**Fig. 2 MUC1 prefers negatively-curved over positively-curved membranes on the same nanostructures. A** Schematic illustration of a nanoX used to induce positive, negative, and zero membrane curvature on the same structure. **B** An SEM image of gradient nanoX arrays with inner angles ranging from 30° (left) to 90° (right), tilted at 45°. All nanoX are 350 nm in arm width, 5 μm in arm length, and 10 μm in spacing. NanoX inner angle (θ) increment: 15°. **C** Confocal images of MUC1_42TR-GFP-transfected U2OS cells cultured on the gradient nanoX arrays. F-actin was stained with phalloidin as a reference. **D** Three sets of zoom-in confocal images show that F-actin prefers the arm ends of nanoXs while MUC1_42TR-GFP prefers the inner faces. Bright-field images of nanoX structures were converted into blue color for visualization purpose. **E** Averaged fluorescence images show the spatial distributions of, F-actin, mCherry-CAAX, MUC1_0TR-GFP, and MUC1_42TR-GFP on nanoXs. **F**, **G** Quantification of end-to-side ratios (reflecting the preference for positive curvature) and inner-to-side ratios (reflecting the preference for negative curvature) of MUC1_42TR-GFP (**F**) and F-actin (**G**) on nanoXs of selected three inner angles. All ratios are normalized against the mCherry-CAAX signals (see Supplementary Table 3A, C for the detailed statistics). **H** An SEM image of a dense nanopillar array at 500-nm-diameter, 1-μm-height, and 1-μm-spacing with a stage tilt of 45°. **I** Confocal images of Hela cells cultured on the dense nanopillar array. MUC1 preferentially locates to inter-pillar spaces where negative membrane curvature can be induced. The bright-field (BF) channel in the merged image is background-subtracted and converted into magenta color for visualization. Scale bars = 10 μm for **B**, **C**, **I**. Scale bars = 2 μm for **H** and zoom-in images in **I**. Welch's *t*-tests (unpaired, two-tailed, not assuming equal variance) are applied for all statistical analyses in this figure. Error bars represent SEM.

as the one shown in Fig. 2H)[37]. For dense nanopillar arrays, the inter-pillar spacing can accommodate or induce the formation of membrane protrusions with negative curvature. The localization of MUC1 at inter-pillar spaces further supports the observation that MUC1 avoids positive membrane curvature induced by nanostructures while preferring negative membrane curvature. A puncta-like distribution of immunostained MUC1 in HeLa cells (Fig. 1K, L and Supplementary Fig. 9C) might also reflect its preference for membrane protrusions with negative curvature.

**MUC1 avoids positively-curved membranes induced by membrane-sculpting proteins**. To examine how MUC1 responds

to biologically-induced membrane curvature, we employed membrane-sculpting proteins, IRSp53, an inverse BAR (I-BAR) protein to induce membrane protrusions with negative curvature (Fig. 3A)[42,43], and FBP17, an F-BAR protein to generate membrane invaginations with positive curvature (Fig. 3B)[43,44]. A truncated FBP17-ΔSH3 mutant (simply denoted as FBP17) was used since it induces membrane invaginations without activating downstream signaling events[45]. U2OS cells usually have few protruding or invaginating structures, as seen in mCherry-CAAX-expressing cells with co-expressing MUC1 variants (Supplementary Fig. 10).

Cells expressing IRSp53-mCherry show dramatic growth of protrusions at the cell periphery, while cells expressing mCherry-

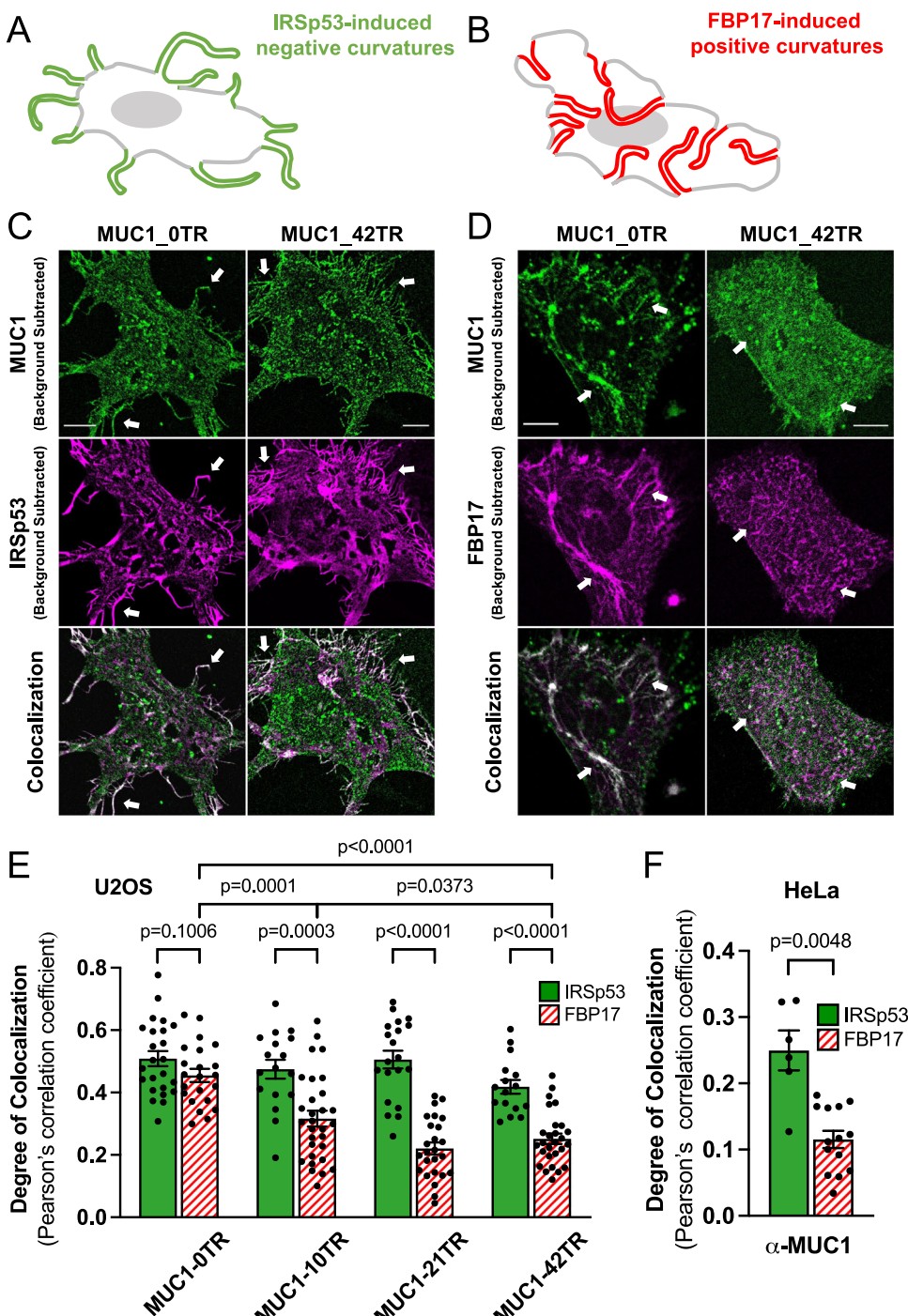

**Fig. 3 MUC1 avoids positively-curved membranes and prefers negatively-curved ones induced by membrane-sculpturing proteins. A**, **B** Schematic illustrations of **A** IRSp53-induced negative membrane curvature and **B** FBP17-induced positive membrane curvature. **C**, **D** Confocal images of U2OS cells transfected with either MUC1_0TR-GFP or MUC1_42TR-GFP and co-transfected with either **C** IRSp53-mCherry to induce membrane protrusions with negative curvature; or **D** mCherry-FBP17 to generate membrane invaginations with positive curvature. Scale bars represent 10 μm. **E** Colocalization analysis of MUC1s of four lengths and two mCherry-BAR-family proteins in U2OS cells shows that longer MUC1s with 21TRs and 42TRs are preferentially excluded from FBP17-induced membrane invaginations (see Supplementary Table 4A, B for the detailed statistics). **F** Colocalization analysis of MUC1 and two mCherry-BAR-family proteins in Hela cells shows a similar trend (see Supplementary Table 4C for the detailed statistics). All cells were cultured on flat surfaces. Welch's *t*-tests (unpaired, two-tailed, not assuming equal variance) are applied for all statistical analyses in this figure. Error bars represent SEM. Arrows were drawn for guidance purpose.

FBP17 show drastic growth of invaginations shown as tubules or puncta in the cell interior. Both MUC1_42TR-GFP and MUC1_0TR-GFP colocalize with the IRSp53-induced membrane protrusions at the cell periphery (Fig. 3C). MUC1_0TR-GFP also colocalizes strongly with FBP17-induced membrane invaginations,

consistent with the pattern of a curvature insensitive membrane marker protein (Fig. 3D). On the other hand, MUC1_42TR-GFP shows little or no overlap with the FBP17-induced membrane invaginations (Fig. 3D). Colocalization was quantified by image registration of two-color channels after background subtraction,

followed by pixel-by-pixel correlation, as reflected by Pearson's correlation coefficient (PCC). Quantifications show that the degree of colocalization between MUC1 and FBP17 is dependent on the length of its ectodomain, and decreases significantly as the number of tandem repeats in MUC1 increases from 0, 10, 21, to 42 (Fig. 3E, Supplementary Table 4A, B, and Supplementary Fig. 11A, B). On the other hand, the colocalization between MUC1 and IRSp53 is not dependent on the ectodomain length as both proteins are present in the protrusions and the flat plasma membranes. We also note that the colocalization between MUC1 and IRSp53 is much less than 100%. This is because unlike CAAX which shows very little intracellular retention, there is always a fraction of MUC1 proteins trapped in the endoplasmic reticulum (ER) or other intracellular organelles, similar to previous reports[40,46]. Nevertheless, there is a significant difference between MUC1 correlations to IRSp53 compared to FBP17—the PCC between MUC1_42TR-GFP and IRSp53 is significantly higher than that between MUC1_42TR-GFP and FBP17.

The C-terminal tail does not affect MUC1's curvature preference. We constructed a full-length MUC1-GFP construct without deleting the C-terminal tail. Similar to MUC1-ΔCT-42TR, full-length MUC1 avoids FBP17-induced membrane invaginations but favors IRSp53-induced membrane protrusions (Supplementary Table 4D and Supplementary Fig. 12A–C). Phalloidin co-staining demonstrates that actin filaments are present in IRSp53-induced membrane protrusions which also have strong MUC1 accumulation, but not in FBP17-induced invaginations (Supplementary Fig. 12D, E).

We further immunostained MUC1 in either IRSp53- or FBP17-transfected HeLa cells. Consistent with our overexpression results and previous findings[22], MUC1 was found to be distributed among the IRSp53-induced membrane protrusions but not the FBP17-induced membrane invaginations (Fig. 3F, Supplementary Table 4C, and Supplementary Fig. 11C, D). The absence of MUC1 on FBP17-induced positive membrane curvature is consistent with the results from the nanostructure experiments. Collectively, these studies using both nanostructure-induced and protein-induced membrane curvatures show that MUC1 avoids positive membrane curvature while preferring negative curvature. This behavior is dependent on the size of its ectodomain.

**Reduction of the glycosylation level reduces MUC1's avoidance of positive curvature.** MUC1 possesses a large number of O-linked glycans, which have been documented to regulate tumor growth/progression[47,48], cell resistance to anoikis[49], immune recognition[50], and clathrin-mediated endocytosis of MUC1 itself[51]. To examine how the glycosylation level of MUC1's ectodomain affects its curvature sensitivity, we made MUC1 mutants of three lengths with reduced O-linked glycosylation based on the previously reported approaches[40]. In each tandem repeat, three out of five serine/threonine O-glycosylation sites were replaced with alanine, leading to a 60% reduction in the O-glycosylation frequency (Fig. 4A, B)[40]. The triple mutants have previously been confirmed to be ~80% less glycosylated than native MUC1 via lectin staining and mass spectroscopy[40].

Using protein-induced membrane curvature, we measured the colocalization of three MUC1 triple mutants (MUC1-T-10TR, MUC1-T-21TR, and MUC1-T-42TR) with IRSp53 or FBP17. Compared with the MUC1s of the same ectodomain length, the mutant MUC1-Ts exhibited a similar degree of colocalization with IRSp53-induced membrane protrusions (Fig. 4C, E, Supplementary Table 4A, and Supplementary Fig. 13A). However, the degree of colocalization with FBP17-induced membrane invaginations is increased for longer (21TR and 42TR) MUC1 triple mutants (Fig. 4D, F, Supplementary Table 4B, and Supplementary Fig. 13B). Similarly, the triple mutant reduces MUC1's avoidance of nanostructure-induced positive curvature. On the 200-nm nanopillar arrays, MUC1-T-42TR triple mutants show higher nanopillar-to-surrounding ratios than native MUC1-42TR (Fig. 4G and Supplementary Table 2B). These measurements indicate that MUC1's avoidance of positive membrane curvature is due to its high level of glycosylation.

We further determined how enzymatic removal of the MUC1 ectodomain affects its curvature preference. We treated MUC1-expressing cells with a mucin-selective protease, the secreted protease of C1 esterase inhibitor from _E. coli_ (StcE)[52]. StcE cleaves the TR domains but not the GFP or the transmembrane domains (Fig. 4H). The loss of the ectodomain entirely removes MUC1's avoidance of nanostructure-induced positive curvature. On nanopillars, StcE treatments increase the MUC1-42TR pillar-to-surrounding ratio to be comparable to the membrane marker CAAX (Fig. 4I and Supplementary Table 2B).

**Large glycoproteins avoid positive membrane curvature in vitro.** To understand whether the curvature preference of MUC1 in cells is due to its intrinsic physical properties or through its interactions with other cellular components, we examined the curvature preference of glycoproteins in vitro on a supported lipid bilayer (Fig. 5A). A recombinant His-tagged and fluorescently-labeled Podocalyxin (Podxl), a mucin-like glycoprotein that is commercially available, was used for this study. The lipid bilayers formed on the gradient nanoX arrays were doped with 30% Ni-NTA-conjugated lipids for recruiting His-tagged Podocalyxin[53] and ~1% rhodamine-labeled lipids for bilayer visualization. It is important to note that for the in vitro experiments where podocalyxin is added onto supported lipid bilayers, the sign of the nanostructure-induced curvature is opposite to that of cellular studies (Fig. 5B). Specifically, curvature at the ends of nanoX mimics membrane protrusions, i.e., negative membrane curvature, while curvature at the inner crosses of nanoX mimics membrane invaginations, i.e., positive membrane curvature.

Fluorescence recovery after photobleaching (FRAP) measurements show that supported lipid bilayers formed on $SiO_2$ nanostructures exhibit similar fluidity to those on flat areas (Supplementary Fig. 14), consistent with previous reports[33,35,54]. Confocal fluorescence imaging of rhodamine lipids shows that the lipid bilayers form relatively uniformly on nanoX arrays while the Podxl signal is more intense at the arm ends of nanoXs (Supplementary Fig. 15A). The Podxl/lipid heatmaps and the intensity ratio quantifications (Fig. 5C, D and Supplementary Table 5A) show that the large glycopolymer exhibits reduced intensities at positively-curved membranes at the center of nanoX, while accumulating more strongly at negatively-curved membranes at the ends of nanoX arms[29,55].

This curvature-dependent podocalyxin distribution only occurs at high protein density. When the percentage of Ni-NTA lipids was reduced to 10%, the podocalyxin distribution on NanoX is similar to that of rhodamine-tagged lipids, indicating a molecular crowding and steric repulsion effect (Fig. 5F, G, Supplementary Table 5B, and Supplementary Fig. 15C). Furthermore, deglycosylation of podocalyxin by glycosidase treatments (Supplementary Fig. 16) largely eliminates the biased distribution of podocalyxin at curved membranes, regardless of the concentration of Ni-NTA lipids (Fig. 5E, H, Supplementary Table 5C, D and Supplementary Fig. 15B, D). Taken together, these in vitro measurements indicate that the avoidance of large glycoproteins to positive curvature is due to its intrinsic physical properties.

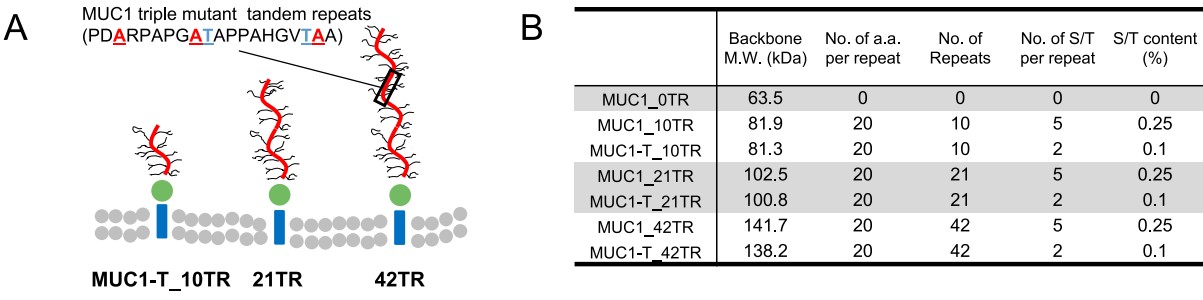

**MUC1's avoidance of positively-curved membranes delays its removal from the plasma membrane**. The generation of positive membrane curvature is an essential feature of endocytosis. We hypothesize that MUC1's avoidance of positive membrane curvature may impede its removal from the plasma membrane through endocytosis. For this purpose, we compared the endocytosis levels of MUC1 with different sizes of ectodomains (MUC1_0TR-GFP, MUC1_10TR-GFP, and MUC1_42TR-GFP). MUC1 endocytosis was probed by first incubating transfected

U2OS cells with anti-GFP under live-cell circumstances at 4 °C, a condition preventing endocytosis. After washing out unbound antibodies, the cultures were serum-starved and warmed to 37 °C to allow endocytosis for a certain time duration. Then surface-bound antibodies were removed by acid stripping before the cultures were immediately fixed, permeabilized, and stained with secondary antibodies for internalized MUC1s (Fig. 6A). For MUC1_0TR-GFP, the internalized anti-GFP signal increases with the time duration at 37 °C from 15, 30, to 60 min, with the signal

**Fig. 4 Reduction of the glycosylation level or cleavage of the ectodomain reduces MUC1's avoidance of positive curvature. A** Schematic illustration of MUC1 triple mutants of varying lengths. **B** Comparison between native MUC1 and triple mutants in their molecular weight and expected glycosylation levels. (MW molecular weight, a.a. amino acids, S/T serine/threonine). **C** Confocal images of U2OS cells co-transfected with IRSp53-mCherry and either MUC1_42TR-GFP or its triple mutant MUC1-T_42TR-GFP. Scale bars represent 10 µm. **D** Confocal images of U2OS cells co-transfected with mCherry-FBP17 and either MUC1_42TR-GFP or its triple mutant MUC1-T_42TR-GFP. Scale bars represent 10 µm. **E** Colocalization analysis of the wildtype and triple mutants of different lengths with IRSp53 (see Supplementary Table 4A for the detailed statistics). **F** Colocalization analysis of the wildtype and triple mutants of different lengths with FBP17 shows that triple mutants of MUC1 with 21TRs and 42TRs have increased colocalization with FBP17 (see Supplementary Table 4B for the detailed statistics). **G** Quantification of nanopillar-to-surrounding ratios shows that triple mutant MUC1-T_42TR-GFP has increased presence at 200-nm nanopillars (see Supplementary Table 2B for the detailed statistics). **H** Schematic illustration of cell surface mucin removal by StcE mucinase. **I** Quantification of nanopillar-to-surrounding ratios shows that StcE treatment significantly increased MUC1_42TR-GFP signals at 200-nm nanopillars (see Supplementary Table 2B for the detailed statistics). All ratios are normalized against the mCherry-CAAX signals. In **C**, **D**, cells were cultured on flat surfaces. Welch's t-tests (unpaired, two-tailed, not assuming equal variance) are applied for all statistical analyses in this figure. Error bars represent SEM. Arrows were drawn for guidance purpose.

intensity at 60 min ~280% of the signal at 15 min (Fig. 6B). On the other hand, for MUC1_42TR-GFP, the internalized anti-GFP signal is much lower and remains relatively unchanged for 15, 30 to 60 min (Fig. 6C). Quantification of the endocytosed signals over many cells confirms that the endocytosis level for MUC1_0TR-GFP is significantly higher than MUC1_42TR-GFP, and the level for MUC1_10TR-GFP lies in between those for 0TR and 42TR (Fig. 6D and Supplementary Table 6A).

The presence of nanopillars (200 nm in diameter, 2-µm in height, and 2.5-µm in spacing) has been shown to enhance the overall cellular endocytosis by inducing positively-curved membranes, which are the hotspots for endocytosis[35]. We found that levels of MUC1_0TR-GFP endocytosis were much higher in nanopillar areas compared to that in flat areas (Fig. 6E and Supplementary Table 6B). However, MUC1_42TR-GFP shows similar low levels of endocytosis on both substrates. These results indicate that the internalization of bulky MUC1_42TR-GFP is not enhanced by positive membrane curvature induced by nanopillars.

Next, we measured how eliminating the MUC1 ectodomain affects its endocytosis efficiency via StcE treatment for MUC1_42TR-GFP. The loss of the ectodomain significantly improves the MUC1 endocytosis level by ~60-100% (Fig. 6F and Supplementary Table 6C). Furthermore, we found that the triple mutant MUC1-T_42TR-GFP with lower glycosylation has ~80–100% more internalization than native MUC1-42TR (Fig. 6F and Supplementary Table 6C). Taken together, even though endocytosis of MUC1 has been previously documented to depend on the cytoplasmic interactions through its C-terminal tail[56], our findings using C-tail-deleted MUC1 indicate that the ectodomain-dependent curvature preference of MUC1 is also a key factor to modulate its endocytosis.

**MUC1's avoidance of positively-curved membranes depends on the curvature value.** To further examine whether MUC1's avoidance of positively-curved membranes depends on the curvature value, we exploited gradient nanobar arrays to induce positive curvature with a range of curvature values (200–2000 nm in width, 1 or 2 µm in length, 1 µm in height, and 5 µm in spacing, Fig. 7A–C). Gradient nanobars allow us to measure many curvature responses from the same culture. As seen by the mCherry-CAAX signal, cell membranes wrap tightly around the nanobars of all widths with slightly higher intensities at the two ends (Fig. 7D and Supplementary Figs. 17, 18). For thin nanobars, in contrast to the membrane marker mCherry-CAAX that shows slightly higher intensity at the ends of nanobars, MUC1_42TR-GFP shows a lower intensity at the ends of nanobars than at the flat side walls (Fig. 7E and Supplementary Fig. 18, white arrowheads). MUC1_42TR-GFP's avoidance of nanobar ends becomes less obvious for wider nanobars (1600 and 2000 nm).

MUC1_0TR-GFP wraps around nanobars of all sizes similar to mCherry-CAAX (Fig. 7F and Supplementary Fig. 18).

We quantified the curvature preference of MUC1s by measuring nanobar end-to-side ratios normalized to mCherry-CAAX ratios. The ratios for MUC1_42TR-GFP are consistently lower than 1 for nanobar widths smaller than 1200 nm, indicating avoidance of positive curvature (Fig. 7G, I). However, the ratios increase to ~1 for nanobar widths of 1600 and 2000 nm. Measurements and quantifications for MUC1_21TR-GFP show a similar trend but a smaller extent of curvature avoidance only for nanobar widths ≤1200 nm (Supplementary Figs. 18, 19B). As the size of MUC1's ectodomain is further reduced, MUC1_10TR-GFP behaves similarly to mCherry-CAAX for the entire range of nanobars (Supplementary Figs. 18, 19A), while MUC1_0TR-GFP shows slightly more enhanced accumulation at the ends of thin nanobars compared to mCherry-CAAX (Fig. 7H, J). StcE-digested MUC1-42TR and MUC1-T_42TR triple mutants showed normalized ratios ~0.97 to 1.05 for the entire range of nanobar width (Supplementary Figs. 18, 19C, D). It is important to note that wider nanobars induce smaller positive curvatures. Through these quantitative measurements, we determine that the curvature range has an upper diameter boundary of ~1200 nm for MUC1_42TR-GFP's avoidance of positive curvature. The upper diameter boundary at 1200 nm is much larger than the diameter of clathrin-coated pits (50–200 nm)[57].

**Discussion**

In this work, we took a multidisciplinary approach to investigate the curvature preference of the glycocalyx protein MUC1. By inducing membrane curvature using either nanostructures or membrane-sculpturing proteins, we found that membrane curvature affects the spatial distribution of MUC1 on the plasma membrane. MUC1 has a reduced presence at positively-curved membranes while it has a preference for negatively-curved ones. Avoidance of positively-curved membranes depends on the length and the glycosylation level of the MUC1's ectodomain. Our in vitro measurements show that MUC1's avoidance of positive curvature is due to steric repulsions of its bulky ecto-domain. Based on these observations, we made an illustration depicting the spatial distribution and the curvature preference of MUC1 at nanotopography- or protein-induced membrane curvature (Fig. 8A, B). Recently, Paszek's group depicted the physical properties of cell-surface mucin polymers by the polymer brush model, in which surface-anchored polymers change their morphology in a density- and size-dependent manner[29,55]. In their model, the end-grafted polymers transit from a mushroom regime to a brush regime at a high surface density to reduce intermolecular steric and electrostatic repulsions. Our observations support this polymer brush model—the avoidance of MUC1 to positively-curved membranes is likely due to intermolecular

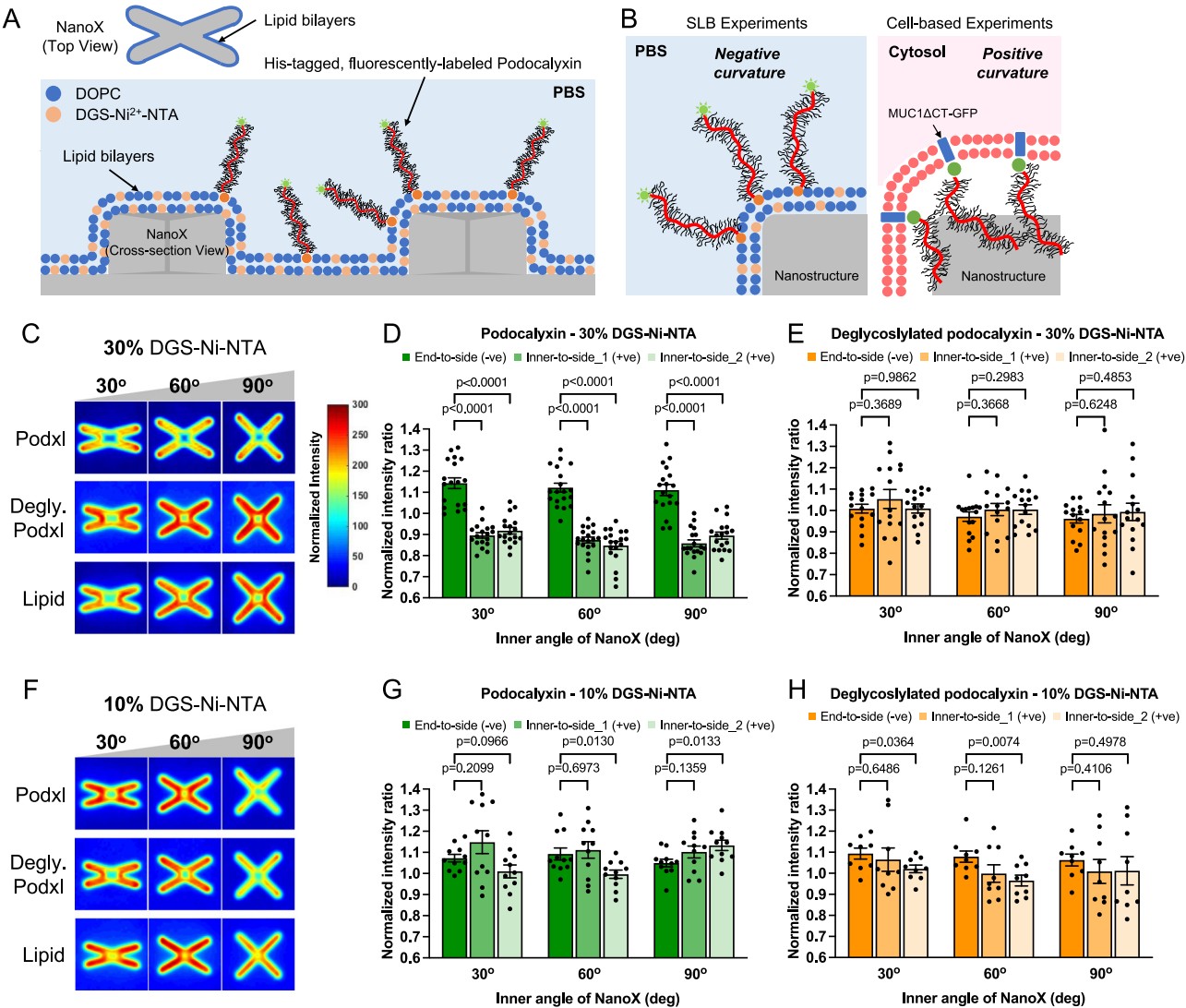

**Fig. 5 Large glycoproteins avoid positive membrane curvatures in vitro. A** Schematic illustration of supported lipid bilayers (SLB)-based arrays employed to study the in vitro curvature preference of a mucin-like glycoprotein, Podocalyxin (Podxl). **B** Cartoons elucidating the opposite membrane curvature for the distribution of mucin glycocalyx proteins on lipid bilayers vs. in the cell-based experiments. **C** Heatmaps show the spatial distributions of Podxl, deglycosylated Podxl, and lipid bilayers around the nanoXs (with 30% DGS-Ni-NTA). **D, E** Quantification of end-to-side (reflecting the preference for negative curvature) and inner-to-side ratios (reflecting the preference for positive curvature) of (**D**) Podxl and (**E**) deglycosylated Podxl on gradient nanoXs arrays of three selected inner angles. Podxl preferentially accumulates at negatively-curved membranes at the ends of nanoX arms. The lipid bilayers were doped with 30% DGS-Ni-NTA (see Supplementary Table 5A, B for the detailed statistics). **F** Heatmaps show the spatial distributions of Podxl, deglycosylated Podxl, and lipid bilayers around the nanoXs (with 10% DGS-Ni-NTA). **G, H** Quantification of end-to-side and inner-to-side ratios of (**G**) Podxl and (**H**) deglycosylated Podxl on the lipid bilayers doped with 10% DGS-Ni-NTA (see Supplementary Table 5C, D for the detailed statistics). All ratios are normalized against the rhodamine-lipid signals. Welch's t-tests (unpaired, two-tailed, not assuming equal variance) are applied for all statistical analyses in this figure. Error bars represent SEM.

repulsions between MUC1's highly glycosylated ectodomain (Fig. 8C, D). Our study indicates that MUC1, similar to IDPs, senses membrane curvature through conformational entropic forces[32].

An interesting consequence of MUC1's avoidance of positive-curved membranes is its reduced endocytosis. Previous studies reported that overexpression of MUC1 does not attenuate endocytosis of other proteins such as transferrin[29]. Therefore, MUC1's avoidance of positive curvature reduces its own internalization, which likely contributes to its long lifetime on the cell surface (20–30 h)[51]. The high density and the long lifetime of MUC1 on the surface of epithelial cells are important for its protective function[12,58]. On the other hand, tumor-associated MUC1s are overexpressed but hypo-glycosylated[13].

Our observations indicate that a reduced glycosylation level of MUC1 is correlated with a reduced avoidance of positive curvature, which makes MUC1 more susceptible to endocytosis. As the intracellular fragment of MUC1 has been shown to be trafficked to the nucleus and modulate transcription factors, the enhanced MUC1 endocytosis of hypo-glycosylated MUC1 in cancer cells may potentiate its oncogenic signaling by increasing its intracellular accumulations[59].

The spatial distribution of bulky glycoproteins on the cell membrane affects how cells interact with extracellular materials. Recent electron microscopy studies of the cell-material interface revealed interesting findings that the gap distance between the cell membrane and protruding nanostructures such as nanopillars (at ~15–20 nm) is much smaller than that between the cell

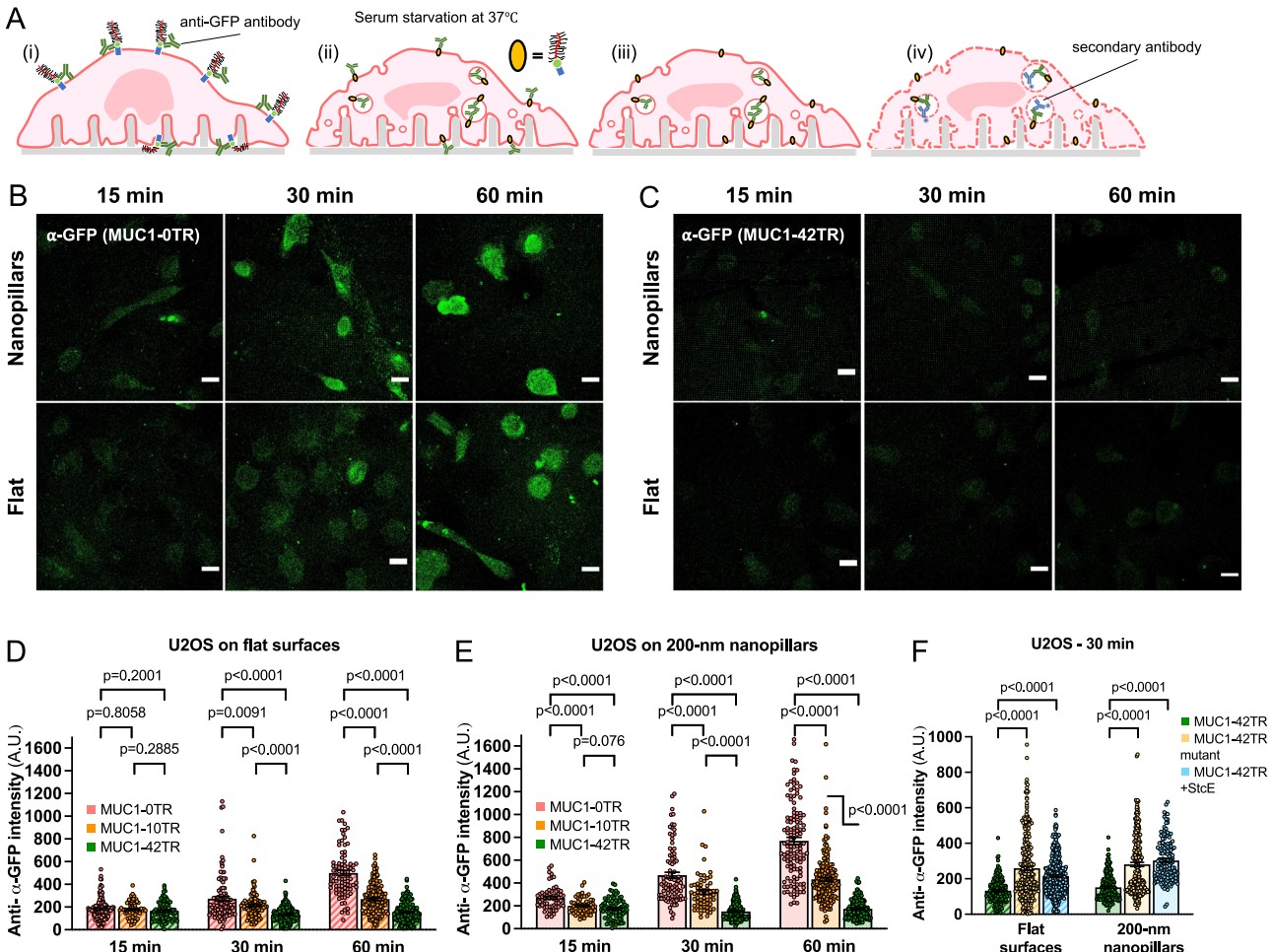

**Fig. 6 MUC1's avoidance of positively-curved membranes reduces its removal from the plasma membrane. A** Schematic illustration of the MUC1 endocytosis assay performed on the 200-nm nanopillar arrays. Cells expressing MUC1-GFP were live-cell stained with anti-GFP antibodies at 4 °C, then warmed up to 37 °C to allow endocytosis for a certain duration under serum-deprived conditions. Then, surface-bound anti-GFP were acid stripped before cells were fixed and probed for internalized MUC1-GFPs. **B**, **C** Confocal images show the immunofluorescent signals of internalized (**B**) MUC1_0TR-GFP and (**C**) MUC1_42TR-GFP in U2OS cells at different time points on either 200-nm nanopillar arrays or flat surfaces. Scale bars represent 20 μm. **D**, **E** Quantifications of MUC1 endocytosis on (**D**) flat surfaces and (**E**) the 200-nm nanopillar arrays. MUC1_42TR-GFP shows significantly reduced endocytosis at 30 and 60 min compared to MUC1_0TR-GFP (see Supplementary Table 6A, B for the detailed statistics). **F** After 30-min incubation, both the triple mutation and the StcE treatment increase the endocytosis level of MUC1_42TR-GFP (see Supplementary Table 6C for the detailed statistics). Welch's $t$-tests (unpaired, two-tailed, not assuming equal variance) are applied for all statistical analyses in this figure.

membrane and flat surfaces (~50–100 nm) or invaginating nanostructures such as nanoholes (>500 nm)[38]. Our finding that bulky glycoprotein avoids positively-curved membranes provides a possible explanation for this phenomenon—the tight association between the cell membrane and nanopillars can only be achieved by excluding these large glycoproteins, often on the order of tens to a hundred nanometers.

## Methods

**Nanostructure fabrication and characterization**. The 200-nm nanopillar, 200-nm nanobar, and gradient nanobar arrays used in this work were fabricated using electron-beam lithography (EBL) and reactive ion etching (RIE) as previously reported[35,36,60]. Briefly, the quartz wafers were spin-coated with electron-beam resist followed by a layer of thin E-Spacer conductive materials. The patterns of nanopillars and nanobars of desired dimensions were inscribed by electron-beam lithography followed by development in xylene. A layer of chromium mask was sputtered on the patterned wafers. The vertical nanostructures were created via RIE. Gradient nanoX-arrayed chips were fabricated in a similar manner as nanopillar and nanobar arrays. Instead, optical photolithography rather than EBL was employed to pattern the wafer. Desired patterns were designed by using an open-source python package. Four-inch quartz wafers were first cleaned with Spin Rinse Dryer (SRD). The cleaned quartz wafers were baked and applied with hexamethyldisilazane (HMDS) to remove residual moisture and promote photoresist adhesion, respectively. Prior to exposure to the desired

pattern of UV using Heidelberg (MLA150), The substrates were coated with a 1-μm-thick photoresist (Shipley 3612). The post-exposure wafers were then immediately subject to baking and development with the MF-26A developer (Transene). A 120-nm-thick layer of chromium mask was deposited on the patterned wafers using an AJA e-beam evaporator and immediately lifted off with acetone and isopropanol. To create nanoX structures, the quartz wafers were etched anisotropically by RIE (Plasma-Therm Versaline LL ICP Dielectric Etcher, PT-Ox) with a mixture of $C_4F_8$, $H_2$, and Ar for 3 min. The substrates were immersed in chromium etchant 1020 (Transene) for 30 min to remove the Cr mask and then incubated in 20:1 Buffered oxide etch (BOE) for 10 min to isotropically shrink nanostructures down to ~350-nm-wide. Nanostructured quartz wafers were then cut into several small chips for biological applications. The shape and dimensions of the nanostructures were measured by scanning electron microscopy (FEI Magellan 400 XHR). Detailed dimensions for different nanostructures are described in the main text.

**Plasmid construction**. MUC1-ΔCT-mOxGFP of varying numbers of tandem repeats (42TR, 21TR, 10TR, and 0TR) are kind gifts from Matthew Paszek Lab at Cornell University[40]. mCherry-CAAX, IRSp53-mCherry, and mCherry-FBP17-ΔSH3 constructs were prepared as previously described[5,43]. Briefly, the DNA fragments coding for the CAAX motif of K-Ras protein (GKKKKKKSKTKCVIM), FBP17-ΔSH3 (a.a. 1–548), or IRSp53 were integrated into the 3′ end (or the 5′ end for IRSp53) of mCherry-encoding vectors. For the MUC1 triple mutant cloning, we exploited Gibson Assembly[61] to generate the plasmids of the MUC1-ΔCT-mOxGFP mutants of three lengths of tandem repeats (10, 21, and 42) using

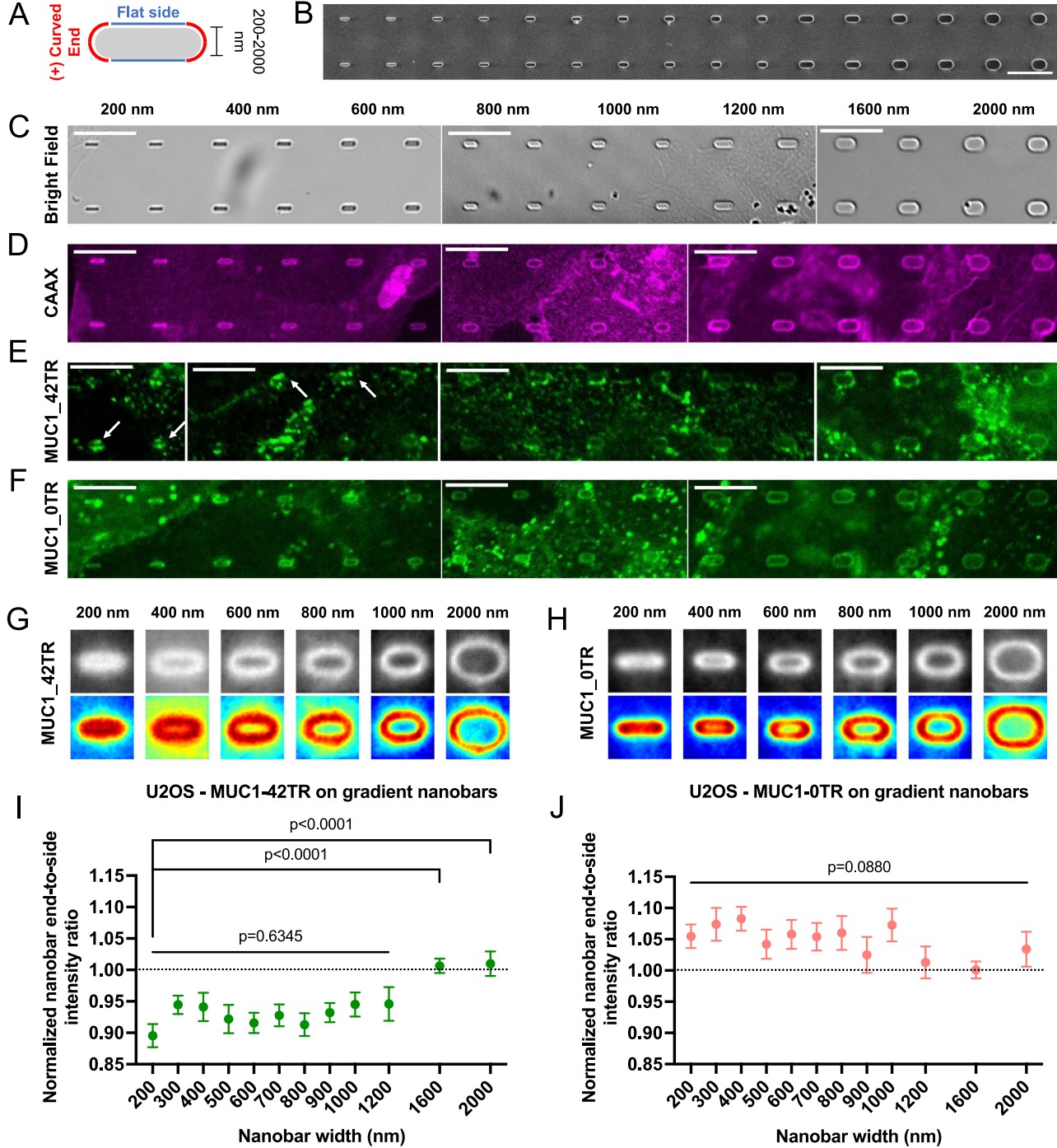

**Fig. 7 MUC1's avoidance of positively-curved membranes depends on the curvature value. A** Schematic illustration of a nanobar inducing both flat and positive curvature. **B** A SEM image and **C** bright-field images of gradient nanobar arrays with widths ranging from 200 (left) to 2000 nm (right). All nanobars are 1 μm in height and 5 μm in spacing. Nanobar width increment: 200 nm for the nanobars smaller than 1200-nm ones and 400 nm for the nanobars larger than 1200-nm ones. **D**–**F** Confocal images of U2OS cells expressing (**D**) mCherry-CAAX, (**E**) MUC1_42TR-GFP, and (**F**) MUC1_0TR-GFP cultured on the gradient nanobar arrays. **G**, **H** Averaged fluorescence images and heatmaps show the spatial distributions of **G** MUC1_42TR-GFP and **H** MUC1_0TR-GFP on the gradient nanobar arrays of six selected widths. **I**, **J** Quantification of **I** MUC1_42TR-GFP and **J** MUC1_0TR-GFP on the gradient nanobar arrays (see Supplementary Table 7 for the detailed statistics). All ratios are normalized against the mCherry-CAAX signals. Scale bars = 10 μm for all images in this figure. Error bars represent SEM. Both Welch's *t*-tests (unpaired, two-tailed, not assuming equal variance) and one-way Welch's ANOVA are applied for the statistical analyses in this figure.

formerly-reported plasmids (pPB_Muc1_0_mOxGFP_dCT_BlpI and pPB_Tet_SumoStar_Muc1_21T_rtTAsM2_IRES_NeoR)[40] as templates. Briefly, the linear vector and the fragment(s) encoding the polymer backbone of triple mutants were amplified using PCR (DNA primer sequences are provided in Supplementary Table 8). Subsequently, the fragment(s) were inserted into the vector and then cyclized in the Gibson Assembly mixture (New England Biolabs, #E2611)

composed of 2 U/μL Taq DNA Ligase, 0.025 U/μL Q5 High Fidelity Polymerase, 0.002 U/μL T5 exonuclease, and 0.05 U/μL DpnI at 50 °C for 30 min.

**Cell plating on the substrates**. Nanostructured chips were cleaned in a piranha solution for 1 h then by air plasma for 20 min. Cleaned nanostructured chips were

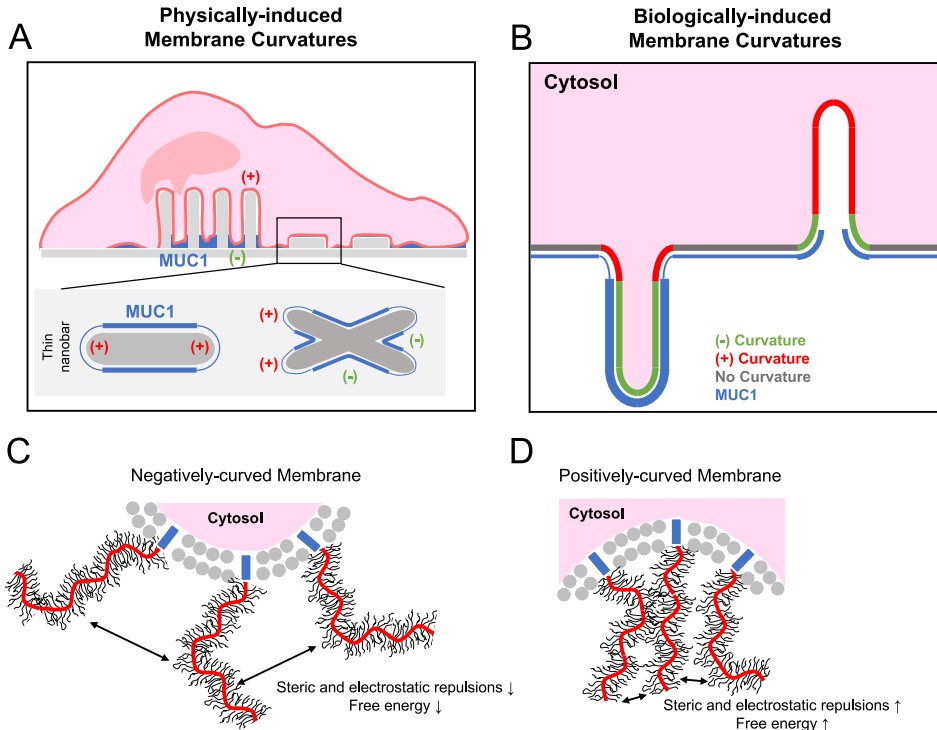

**Fig. 8 Proposed models of how bulky glycoproteins respond to membrane curvature. A** On vertical nanostructures, MUC1 shows a preference for negatively-curved and flat membranes over positively-curved ones. **B** MUC1 prefers IRSp53-induced negatively-curved membranes rather than FBP17-induced positive curvature. **C** Molecular repulsions and the free energy of bulky glycoproteins are minimized as the cell membrane accommodates a negative curvature. **D** Steric and electrostatic repulsions and the free energy are increased when the cell membrane adopts a positive curvature.

placed into 12- or 24-well plate, then coated with 0.2 mg/mL poly-L-lysine (PLL) (Sigma-Aldrich, #P5899) for 20 min, 0.5% glutaraldehyde (Sigma-Aldrich, #354400) for 20 min, and 1:1 (0.02 mg/mL) mixture of fibronectin (Sigma-Aldrich, #F1141) and gelatin (Sigma-Aldrich, #G9391) for 15 min. After that, the substrates were disinfected with 70% ethanol for 30 min and then incubated with 1X DMEM (Gibco)(with 10% fetal bovine serum (FBS)(Cytiva, #SH30071) but no antibiotics) for 30 min to quench free aldehydes. All processes were carried out at room temperature. Desired amounts of either U2OS (ATCC, HTB-96) or Hela (ATCC, CCL-2) cells (un-transfected or transfected) were then cultured on fibronectin/gelatin-coated nanostructured substrates and maintained in the DMEM medium supplemented with 10% FBS, 100 U/mL penicillin and 100 mg/mL streptomycin (Gibco, #15140122). The cultures were incubated in a standard incubator for 24–48 h at 37 °C with 5% $CO_2$. For the experiments involving BAR-family proteins, a flat 22 mm × 50 mm glass coverslip was sterilized and then sealed with custom-made polydimethylsiloxane (PDMS) well. The coverslip was subsequently incubated with a 1:1 (0.02 mg/mL) mixture of fibronectin and gelatin for 2 h at 37 °C. After three washes with 1X phosphate-buffered saline (PBS) (Gibco), desired amounts of MUC1/IRSp53- or MUC1/FBP17-co-transfected cells were cultured on the PDMS well-sealed glass coverslip and incubated under the aforementioned condition.

**Transfection using electroporation**. Confluent U2OS or HeLa cells were first trypsinized for 5–15 min at 37 °C to detach cells. After spinning by centrifuge, aspirating the supernatant and resuspending with the growth medium (1X DMEM without antibiotics). Second centrifugation was employed to completely remove residual trypsin and EDTA. Plasmid DNA of interest of a given concentration (~1.5 μg for MUC1-ΔCT-GFP and triple mutants of varying lengths; ~0.3 μg for mCherry-CAAX; ~1.3 μg for IRSp53-mCherry and mCherry-FBP17-ΔSH3) were mixed in a solution composed of 2 μL of electroporation buffer I (360 mM adenosine 5′-triphosphate and 600 mM magnesium chloride) and 100 μL of electroporation buffer II (88 mM monobasic potassium phosphate and 14 mM sodium bicarbonate at pH = 7.4). After removing the supernatant, cells were gently mixed with the electroporation mixture and subsequently transferred into a 0.2-cm electroporation cuvette (Thermo Fisher Scientific, #FB102). The cell-DNA mixture was then electroporated with a cell type-specific program using Amaxa Nucleofector II (Lonza). To complete the electroporation, the mixture was immediately added with 650 μL of pre-warmed (37 °C) 1X DMEM (without antibiotics) and incubated for 5 min. After spinning and resuspending with 500 μL of growth medium (1X DMEM with 10% FBS and antibiotics), the transfected cells at 1:8 dilution were plated on the cleaned nanostructured chips and cultured at 37 °C for 24–48 h.

**Cell membrane visualization**. To visualize the plasma membrane via confocal imaging, we applied two methods in this study: (1) (For HeLa cells) Membrane chemical staining with a lipid-soluble dye (CellMask Orange or Green Plasma membrane Stain, Invitrogen, #C10045 and #C37608); and (2) (For U2OS cells) Transient expression of fluorescent CAAX (FusionRed-CAAX or mCherry-CAAX)[41,62]. For CellMask staining, HeLa cells were cultured overnight on nanostructured substrates and then treated with 0.25 μg/mL CellMask for 5 min at 37 °C prior to MUC1 immunostaining, fixation, and permeabilization. Cells were subsequently washed with cold 1X PBS to prevent dye internalization. For CAAX expression, overnight-cultured U2OS cells were co-transfected with mCherry-CAAX or FR-CAAX via electroporation.

**Immunostaining and counterstaining**. AP2 (in both U2OS and HeLa cells) and MUC1 on Hela cells were visualized by immunostaining. To stain AP2, overnight-cultured cells were fixed in 4% paraformaldehyde (Thermo Fisher Scientific, #28908) at room temperature for 20 min. Cells were then permeabilized with 0.1% Triton X-100 (Sigma-Aldrich Corporation, #T8787) in 1X PBS for 5 min after three washes with 1X PBS. Subsequently, the cell-coated nanostructured substrates were blocked with 1% bovine serum albumin (BSA)(Sigma-Aldrich, #A9418) and 0.1% Triton X-100 in 1X PBS (staining buffer) at room temperature for 30 min. Cells were then incubated with mouse Anti-alpha Adaptin antibody [AP6] (anti-AP2, Abcam, #ab2730) at 1:500 dilution in the staining buffer for 2–24 h at 4 °C. After three washes with 1X PBS, the samples were then stained with secondary antibodies (goat anti-mouse IgG Alexa Fluor 488, 594, or 647, Invitrogen, #A-11029, #A-11032, and #A-21236) at 1:1000 dilution in the staining buffer for 1 h at room temperature in the dark. In the same step, Hoechst dye (Thermo Fisher Scientific, #62249) and phalloidin (Alexa Fluor 594 or 647-labeled, Invitrogen, #A12381 and #A22287) were added to stain nuclei and F-actin, respectively. Cells were washed with 1X PBS three times prior to confocal imaging. For MUC1 immunostaining on Hela cells, live HeLa cells were incubated with mouse anti-MUC1/episialin antibody (clone 214D4, Sigma-Aldrich, #05-652) at 1:500 dilution in cold 1X PBS for 2 h at 4 °C. After three gentle washes with cold 1X PBS, the samples were then stained with secondary antibodies (goat anti-mouse IgG Alexa Fluor 488, 594, or 647, Invitrogen) for 1 h at 4 °C in the dark. Cells were subsequently washed with cold 1X PBS gently three times prior to fixation, permeabilization, and confocal imaging.

**Flow cytometric analysis**. MUC1-ΔCT-expressing U2OS cells were detached non-enzymatically with a cell dissociation buffer (Gibco, #13151014) at 37 °C for 15–20 min. After three gentle washes with an ice-cold FACS buffer (0.5% BSA in

1X PBS), cells were stained with rabbit anti-GFP antibody (Sigma-Aldrich, #PC408) at 1:500 dilution in the FACS buffer for 30 min on ice. After washing with the ice-cold FACS buffer, cells were then labeled with goat anti-rabbit IgG Alexa Fluor 647 (Invitrogen, #A-32733) at 1:1000 dilution for 30 min on ice in the dark. Free antibodies were then removed by an ice-cold 2 mM EDTA in the FACS buffer. Sytox Blue (Invitrogen, #S34857) was then added to cells to check cell viability. A MACSQuant flow cytometer (Miltenyi Biotec) was used for the analysis. The raw data were further processed using FCS Express™ 7 (De Novo Software).

**StcE mucinase treatment**. MUC1-42TR-ΔCT-expressing U2OS cells were treated with 25 μg/mL StcE mucinase for 30 min at 37 °C to digest cell surface mucins. Higher enzyme concentration (5X higher than the suggested one)[52] and shorter incubation time could minimize internalization caused by the loss of cell surface mucins. After three washes with 1X PBS, the StcE-treated cells were subject to fixation, permeabilization, and staining.

**Supported lipid bilayer (SLB) experiments on the gradient nanoX arrays**. The supported lipid bilayers are composed of 70 mol.% 1,2-dioleoyl-*sn*-glycero-3-phosphocholine (DOPC), 30 mol.% 1,2-dioleoyl-*sn*-glycero-3-[(*N*-(5-amino-1-carboxypentyl)iminodiacetic acid)succinyl](nickel salt) (DGS-Ni²⁺-NTA) (or 90 mol.% DOPC + 10 mol.% DGS-Ni²⁺-NTA)(Avanti Polar Lipids, #850375 C and #790404 C) and doped with ~1 mol.% of Texas Red-1,2-dihexadecanoyl-*sn*-glycero-3-phosphoethanolamine (DHPE-TXR)(Invitrogen, #T1395MP), were prepared in a solvent-assisted (SALB) manner[63]. Briefly, lipids were dissolved and mixed in chloroform. The mixture was then dried in a clean glass vial with nitrogen and desiccated for 30 min. The 0.5 mg of dried lipids were dissolved in IPA to make a 0.5 mg/mL lipid mixture followed by sonication at room temperature for 5–10 min. The piranha- and plasma-cleaned nanostructured substrates were first sealed with custom-made PDMS wells followed by incubation with IPA for 5 min. Subsequently, IPA was slowly exchanged with 50 μL of 0.5 mg/mL lipid mixture using microliter syringes. After a 5-min incubation, the lipid-IPA mixture was exchanged with 1X PBS slowly and gently using microliter syringes for 10 min to complete the solvent-assisted lipid bilayer (SALB) formation procedure. After a 5-min wait, the mixture was gently washed with 1X PBS to remove residual vesicles. Approximately 2 μM Alexa Fluor 647-labeled recombinant His-tagged Podocalyxin (native or deglycosylated) in 1X PBS was then added to the SLB-coated nanostructured substrates and incubated for ~45 min at 37 °C. Before confocal imaging, the samples were gently washed with 1X PBS five to six times.

**Fluorescence recovery after photobleaching (FRAP)**. For FRAP assay, a region of interest on SLB-coated nanoX arrays was selected and bleached with laser at 594 nm for 10 s. After photobleaching, the lipid bilayer was imaged at a frequency of 0.25 Hz for 200 s using an epi-fluorescence microscope (Leica DMI 6000B). The fluorescence intensities in the bleached area were measured using Fiji (ImageJ2, version 2.3.0). The intensity was then normalized to that in the unbleached regions.

**Recombinant His-tagged Podocalyxin modifications**. Recombinant His-tagged human Podocalyxin protein (R&D Biosystems, #1658-PD-050) was fluorescently labeled and deglycosylated as needed. About 1 mg/mL of His-tagged Podocalyxin was mixed with 25 molar equivalent of Alexa Fluor 647 NHS ester (*N*-hydroxysuccinimidyl ester) (Thermo Fisher Scientific, #A37573) and incubated for 2 h at room temperature. Free fluorophores were removed using 40 kDa 0.5 mL Zeba columns. To deglycosylate His-tagged Podocalyxin for control experiments, 2.5 mg/mL of His-tagged Podocalyxin was mixed with Protein Deglycosylation Mix II (New England Biolabs, #P6044S) and incubated for 30 min at room temperature. Subsequently, the reaction was transferred to 37 °C and incubated for an extra 16 h. The degree of labeling was determined spectrophotometrically; The degree of deglycosylation and labeling were further confirmed via gel electrophoresis and fluorescence gel imaging at the excitation wavelength of 700 nm (Supplementary Fig. 16).

**MUC1 endocytosis experiments**. MUC1-ΔCT-GFP (0, 10, 42TR, or 42TR triple mutants)-transfected U2OS cells were plated on fibronectin/gelatin-coated 200-nm nanopillar chips (200-nm in diameter, 2-μm in height, and 2.5-μm in spacing) and cultured in the DMEM medium supplemented with 10% FBS and antibiotics at 37 °C with 5% CO₂. After 24–48 h incubation, cells were stained with rabbit anti-GFPat 1:500 dilution in ice-cold 1X PBS for 1–1.5 h at 4 °C. After three gentle washes with cold 1X PBS to remove free anti-GFP antibodies, cells were treated with pre-warmed serum-free growth medium (1X DMEM; without FBS and antibiotics) to initiate endocytosis and incubated at 37 °C for desired periods of time. For the mucinase experiments, StcE was included in the pre-warmed serum-free growth medium. Afterward, the internalization was stopped by rapidly cooling cells on ice followed by three gentle washes with cold 1X PBS. To strip away un-internalized anti-GFP antibodies, cells were incubated with an acid buffer (100 mM glycine, 150 mM NaCl, pH = 2.2) for a minute. After three 1X PBS washes, cells were fixed and permeabilized at room temperature. After three washes with 1X PBS, the samples were then stained with goat anti-rabbit IgG Alexa Fluor Plus 647 at 1:1000 dilution in the staining buffer for 1 h at room temperature in the dark before confocal imaging. MUC1 endocytosis efficiency was reflected by the temporal change of the fluorescence intensity of internalized anti- anti-GFP antibody signals.

**Confocal imaging**. All fluorescence images (except for FRAP images) were acquired using the Nikon A1plus confocal microscope with a 60X oil immersion objective (NA = 1.4). When imaging fixed cells or supported lipid bilayers on nanostructured chips, we flipped and placed the nanochips in a glass-bottom Petri dish since the 60X objective has a short working distance[36]. For imaging IRSp53- or FBP17-co-transfected cells cultured on a flat glass surface, the cell-coated coverslip was flipped, mounted, and glued on a 25 mm × 75 mm × 1 mm microscope slide (Thermo Fisher Scientific, #12-550-15) after fixation, permeabilization, and staining. MUC1-ΔCT-GFP, mCherry-tagged IRSp53, FBP17-ΔSH3, and CAAX were imaged at the excitation wavelengths of 488 and 561 nm, respectively. Immunostained or counterstained cellular components were imaged at the corresponding excitation wavelengths.

**Quantification of the fluorescence signals of proteins on the nanostructured substrates**. Confocal images were processed and analyzed using MATLAB (2018a) and Fiji (ImageJ2, version 2.3.0). The quantification was performed by averaging many nanostructures obtained from the transfection and immunostaining experiments. We use a custom-written MATLAB code[36] to create a matrix of masks covering each nanostructure of identical dimension in the bright-field channel. These masks were then applied to the corresponding fluorescence images (for example, GFP channel for MUC1-ΔCT, mCherry channel for CAAX, etc.) to create the average image. The heatmaps were calculated using a custom-written MATLAB code. The average fluorescence images were then processed by Fiji (ImageJ2, version 2.3.0) to obtain nanobar end-to-side or nanopillar-to-surrounding ratios. Based on the integrity of the nanostructures and the position of a cell adhered to nanostructured substrates, a single cell may cover a wide range of the numbers of nanostructures from ~15 nanostructure/cell to ~250 nanostructure/cell. See Supplementary Fig. 2 for the detailed description and exemplary process flowchart for analysis.

**Quantification of the degree of colocalization between MUC1-ΔCT-GFP and BAR-family proteins**. For the colocalization analysis, confocal images were processed and analyzed using ImageJ. Prior to quantitative analysis, the regions of interest (ROIs) were created to select individual cells for analysis. Both green (MUC1-ΔCT-GFP and triple mutants of varying lengths or MUC1 immunostained with Alexa Fluor 488) and red channels (IRSp53-mCherry or mCherry-FBP17-ΔSH3) of the cell images were subject to background subtraction using rolling ball algorithm with a radius of 10–20 pixels. The level of colocalization between two channels, reflected by Pearson's correlation coefficient (PCC), were analyzed using the plug-in in Fiji (ImageJ2, version 2.3.0).

**Statistics and reproducibility**. Welch's *t*-tests (unpaired, two-tailed, not assuming equal variance) was employed to evaluate the statistical significance (Figs. 1J, N, O; 2F, G; 3E, F; 4E–G, I; 5D, E, G, H; 6D–F; 7I, J and Supplementary Figs. 4C; 5E; 8; 12C; 19). For Fig. 7I, J and Supplementary Fig. 19, we also applied one-way Welch's ANOVA to compare more than two groups. Statistical significance is considered when $p < 0.05$. All data were presented as mean ± SEM (SD/$\sqrt{N}$; $N$ = number of microscopic fields of view or number of cells) or mean ± SD as mentioned in the figure captions. Detailed statistics for each quantification graph are shown in Supplementary Tables 1–7. On average, there are 1–3 cells per microscopic field of view. All statistical analyses were performed using Prism 9 (GraphPad Software). All nanostructure-involved experiments were repeated three to five times; BAR-family protein-involved experiments and endocytosis experiments were repeated at least two to three times. All measurements were taken from distinct samples, no sample was measured repeatedly.

**Reporting summary**. Further information on research design is available in the Nature Research Reporting Summary linked to this article.

## Data availability
All data that support this work are included in the main figures, supplementary information, and Source Data.

## Code availability
Matlab scripts for image analysis on nanostructured arrays are available from the corresponding authors upon request.

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

## Acknowledgements
We thank Ms. Hao Pan and Prof. Matthew J. Paszek from Cornell University for the kind gift of MUC1△CT-mOxGFP constructs; Dr. Zeinab Jehad from the B.C. group at Stanford Chemistry for advice on the nanostructure fabrication; Ms. Judy Shon from the C.R.B. group at Stanford Chemistry for the kind gift of the full-length MUC1-mOxGFP plasmid; Ms. Gabby Tender from the C.R.B. group for the kind assistance in flow cytometric experiments; Nanofabrication and SEM characterization of vertical nanostructures were performed in Stanford Nanofabrication Facility (SNF) and Stanford Nano Shared Facilities (SNSF). This work was supported by the National Institutes of Health (R35GM141598 to B.C.), Stanford University Center for Molecular Analysis and Design (CMAD) fellowship (to C.-H.L.).

## Author contributions
C.-H.L., K.P., C.R.B., and B.C. conceptualized and designed the research; C.-H.L. performed all cell-based and supported lipid bilayer experiments and analyzed all data. X.L. and C.-T.T. designed and fabricated the gradient nanobar arrays. C.-T.T. designed the gradient nanoX arrays. C.-T.T. and C.-H.L. fabricated the gradient nanoX arrays. X.L. and C.-T.T. took SEM images for the nanostructured substrates. K.P. purified StcE mucinase, labeled and deglycosylated recombinant Podocalyxin. T.J. and C.-H.L. made MUC1 triple mutant constructs. B.C. and X.L. developed the Matlab code for the analysis. C.-H.L. and M.L.N. imaged HeLa cells on dense nanopillars. C.-H.L. and B.C. wrote the paper. All the authors discussed the results and commented on the manuscript.

## Competing interests
Two contributing authors declare the following competing interests: K.P. and C.R.B. are listed as inventors on a patent application filed by Stanford University relating to the use of enzymes to digest mucin-domain glycoproteins (WO2020097386A1). C.R.B. is a cofounder and scientific advisory board member of Lycia Therapeutics, Palleon Pharmaceuticals, Enable Bioscience, Redwood Biosciences (a subsidiary of Catalent), OliLux Bio, Grace Science LLC, and InterVenn Biosciences. The remaining authors declare no competing interests.
