## [Peer Review File · Nature Communications]

Membrane curvature regulates the spatial distribution of bulky glycoproteinsREVIEWER COMMENTS

Reviewer #1 (Remarks to the Author):

The MS by Lu et al. describes the ability of MUC1 (one of the major components of the mammalian glycocalyx) to sense the curvature of the plasma membrane and prefer/avoid areas of respectively -ve/+ve curvature. MUC1's curvature preference is suggested to reduce its endocytic uptake.

Overall the MS is very well written, and referenced, and the figures are very well presented. Mechanistically MUC1-curvature sensing appears to bare strong similarities with IDPs as investigated by Stachowiak et al. They have demonstrated in two NatComm papers first that IDPs help induce curvature and then that they can sense it. It appears that the curvature inducing properties of MUC1 have already been documented. One could claim that curvature sensing by MUC1 is rather predictable, and not that exciting at least from a biophysical perspective. However, it would appear that there is sufficient biological interest in demonstrating curvature sensing by MUC1 and by extension glycoproteins, as e.g. in the case of endocytosis. However, as I explain below in detail, I have major concerns on whether many of the claims in this MS are supported by the data, thus I am unfortunately not able to recommend publication.

1. Claim: "On the same nanobars, MUC1_42TR-GFP appears to distribute more on the flat side walls (Fig. 1K)". Upon close inspection of Fig. 1K and especially of the averages in Fig. 1M, I see uniform distribution for 21TR and 42TR. Which appears to be in contradiction with the analysis shown in Fig. 1N. The authors should comment on this apparent discrepancy.

2. Claim: "In contrast, on the same nanoXs, MUC1_42TR-GFP accumulated more at the inner faces with negative curvatures. ". After close inspection of Fig. 2C-D, and of the averages in 2E/S5, I am not able to confirm claim about 42TR. Actually, in S5 intensities appear higher on the edges. Which appears to be in contradiction with the analysis shown in

Fig. 2F. The authors should comment on this apparent discrepancy..

3. In the experiments with I-, F-bars, according to the way the authors have interpreted the results from fig. 1 and 2, one would expect that truncation would produce an antisymmetric response (up/down concentration) to the +/- curvatures. Such an antisymmetric response

21

Figure 2. MUC1 prefers negatively-curved over positively-curved membranes on the same nanostructures.

(A) Schematic illustration of a nanoX used to induce positive, negative and zero membrane curvatures on the same structure. (B) An SEM image of gradient nanoX arrays with inner angles ranging from 30° (left) to 90° (right). All nanoX are 350 nm in arm width, 5 μm in arm width and 10 μm in spacing. NanoX Inner angle (θ) increment: 15°

. Scale bar: 10 μm. (C) Confocal images of

MUC1_42TR-GFP-transfected U2OS cells cultured on the gradient nanoX arrays. F-actin was stained with phalloidin as a reference. Scale bar: 10 μm. (D) Three sets of zoom-in confocal images show that F-actin prefers the ends of nanoXs while MUC1_42TR-GFP prefers the inner faces. Bright field images of nanoX structures were converted into blue color for visualization purposes. (E) Averaged fluorescence images show the spatial distributions of, F-actin, mCherry-CAAX, MUC1_0TR-GFP and MUC1_42TR-GFP on nanoXs. (F-G) Quantification of end-to-side ratios (reflecting positive curvature) and inner-to-side ratios (reflecting negative curvature) of MUC1_42TR-GFP (F) and F-actin (G) on nanoXs of selected 3 inner angles. All ratios are normalized against the mCherry-CAAX signals. (H) An SEM image of a dense

23

Supplementary Figure S5. Heatmaps depicting the intensity distribution of mCherry-CAAX, two MUC1-ΔCT-GFP, and F-actin signals on U2OS cells plated on the gradient nanoX arrays (Cell-based experiments).

All nanoX are 350 nm in width and 10 μm in spacing. NanoX Inner angle (θ) increment: 15°

.

6

would validate the mechanism. However, the data show that the colocalization of MUC1 and I-BAR is not at all affected by the length of the ectodomain. Similarly, the triple mutation did not affect the colocalization of MUC1-Ts with IRSp53. To the same degree that the experiments with the F-BAR validate the mechanism, the lack of response in the I-BAR experiments refutes it. This is a significant problem that is neither acknowledged nor addressed, let alone resolved.

4. Claim: “In sharp contrast, MUC1_42TR-GFP prefers the flat sidewalls of thin nanobars but avoids the two ends with positive curvatures (Fig. 7E).” I do not think that the data I see in 7E or in 7G support this claim. This appears to be in contradiction with the analysis shown in Fig. 7I. The authors should comment on this apparent discrepancy.

5. Claim: “Through these quantitative measurements, we determined the curvature threshold, or the cutoff value, to be $\sim 1/1200$ nm for MUC1_42TR-GFP’s avoidance of positive curvatures. The threshold diameter at 1200 nm is much larger than the diameter of clathrin-coated pits (50-200 nm).” My understanding of the way entropic forces-based curvature sensing works is based on the data from the Stachowiak group on IDPs. There they do not see, nor expect to see, thresholds but rather 1) monotonic dependencies that 2) get steeper as curvatures increase. Both of these features appear to contradict the findings of Fig. 7 suggesting a different mechanism. The authors should comment on this apparent discrepancy.

6. Claim: “For this purpose, we designed and fabricated the gradient “NanoX” arrays which are able to induce positive curvatures at the arm ends and negative curvatures at the inner faces (Fig. 2A).” The authors should provide support for the claim that membrane conforms to the X shapes.

7. Claim: “For dense nanopillar arrays, Fig. 2I, the inter-pillar spacing was small enough to induce the formation of membrane protrusions with negative curvatures in the inter-pillar

spaces. ” The authors should provide data to support this claim.

8. Claim: “Fluorescence recovery after photobleaching (FRAP) measurement shows that supported lipid bilayers formed on SiO₂ nanostructures exhibited similar fluidity as on flat areas (Suppl. Fig. S12)”. S12 does not compare FRAP curves on the two areas and thus does not support this claim.

General comments:

Figure 7. MUC1’s avoidance to positively-curved membranes depends on the curvature value.

(A) Schematic illustration of a nanobar inducing both flat and positive membrane curvatures. (B) A SEM image of gradient nanobar arrays with the widths ranging from 200 (left) to 2000 nm (right). All nanobars are 1 μ m in height and 5 μ m in spacing. Nanobar width increment: 200 nm for the nanobars smaller than 1200-nm ones and 400 nm for the nanobars larger than 1200-nm ones. Scale bar: 10 μ m. (C) Confocal images of U2OS cells expressing mCherry-CAAX, MUC1_42TR-GFP and MUC1_OTR-GFP cultured on the gradient nanobar arrays. Scale bars represent 10 μ m. (G-H) Averaged fluorescence images and heatmaps show the spatial

31

The prevailing natural phenotype in the spatial distribution of MUC1, as seen in all cell figures in this MS, appear to be puncta. The authors should comment on this and if possible link them to curvature.

Work with IDPs would suggest that surface density, i.e expression levels, strongly influence the coupling to curvature. Have the authors quantified the expression levels of the different mutants and if so what where they, and where they the same.

It would be very helpful to provide info on the number of different cells and numbers of pillar used on the averages, in addition to the number of independent repeats that should be ≥ 3 in all data sets (especially when SEM is used to reduce s.d.).

Reviewer #2 (Remarks to the Author):

The manuscript titled: "Membrane curvature regulates the spatial distribution of bulky glycoproteins" by Lu et al examines the biophysical behavior of MUC1, a bulky glycoprotein implicated in a range of important biological functions, in curved membranes. The authors used a powerful combination of nanoarray technologies and molecular biology to enforce curvatures on cell membranes with tunable geometries to evaluate how perturbations in membrane morphology affects MUC1 distribution in the membranes.

This work builds on a recent discovery that bulky mucins have the capacity to enforce curvature in membranes and this work shows that this relationship is reciprocal and that membrane curvature may likewise lead to reorganization of the mucinous glycocalyx, which can have biological consequences, such as increased resistance of mucins to enter endosomes leading to their prolonged cell surface residence.

The key takeaways from this study are the findings that bulky mucins avoid membranes with negative curvatures and that this propensity increases with the bulk of the mucin ectodomain (as defined by its length and level of glycosylation) and membrane density (shown in supported lipid bilayers).

The methods are well designed and the data are clearly presented. All appropriate controls are included and the manuscript is very well written and easy to follow.

The paper provides important new insights into the mechanobiology of the mucinous glycocalyx and I recommend it is suitable for publication as is.

I only had two minor comments for the authors to consider.

1) In certain instances, I found it difficult to visually confirm the changes in Muc1 localization on the nanopillar nanobar arrays using the fluorescence micrographs in Figures 1 and 2. For instance in Fig 1 K an M, I was having a hard time observing the avoidance of the positively curved ends of the nanobar by MUC1-42TR. The distribution of the protein appears quite similar to that of CAAX. Similarly, the distributions of MUC1 in Fig 2 D and E are quite difficult to assess visually (the additional SI heatmaps help). I understand that the authors applied a custom MATLAB program to quantify changes in fluorescence. Perhaps a description of how that process works in the SI would be helpful.

2) My second comment relates to the role of the cytosolic domain of Muc1 in the membrane localization process. The authors comment on its role in endocytosis and signaling. I wonder to what extent the

interactions of the cytosolic domain with the cytoskeleton may counteract the avoidance of positive curvatures by MUCs. Or can the avoidance of such membrane regions drive the reorganization of the cytoskeleton?

Reviewer #3 (Remarks to the Author):

The manuscript "Membrane curvature regulates the spatial distribution of bulky glycoproteins" by Lu et al. describes a series of experiments conducted to look out for the influence of membrane curvature on the glycoprotein MUC1.

The manuscript is overall well written and organized. The authors proceed in a clear structure for the different experiments to consecutively follow logically a path to evidence the proposed hypothesis on how the membrane curvature impacts the distribution of MUC1. The work in the experiments is substantial and makes the idea of physical crowding / space requirements being the driving force for the observation that MUC1 avoids positively curved membrane areas.

The only criticism I have is about the presentation and discussion of the entanglement of whether glycoproteins cause membrane curvature (as shown by the works referenced in the second abstract of the introduction) or are guided by membrane curvature, or more likely what the extent of both effects is. I think the best evidence for the "sensing not sculpturing" is given by the author also adding the in vitro experiments, as there a very simple system without any other components and proteins that could play a role in addition to the MUC1 is presented. So in this context, I was feeling a bit confused when after the the second paragraph (with the references to work showing glycoproteins can cause membrane curvature), the next paragraph first discussed generally about the role of membrane curvature for cell physiology and only then comes back to the contrast "curvature sculpture against sensing" again. So I would suggest to move the general introduction to the role of membrane curvature (the passage "Cell membranes are dynamic [...] cell invasion") to another place, maybe directly after the introduction on MUC1 or even as the overall start of the whole introduction, to have the contrasting section ("Although it is now established [...]") directly following on the introduction of glycoproteins sculpturing curvature. This would in my opinion even more clearly present the main point and novelty of the authors work.

One question to the fluorescence images in Figure 1c. Looking at the a-MUC1 / CellMask channels, it looks like there is a kind of "window" in the cell with overall reduced fluorescence intensity, as only in this region the raised CellMask signal on/near the pillars is clearly visible (while the rest looks more homogeneously red). Additionally, the a-MUC1 signal is raised in these areas to a point where the pillars

become much more visible (e.g. in the area right to the dash marked box). Have the authors some idea for what this effect is?

And (also Figure 1c), for further clarity, I suggest that the authors include a full set (a-AP2, CellMask, Bright field) of images also for the separate experiment in the fourth panel (a-AP2) - can be done in the supporting information of course to not clutter the Figure 1 even more.

Having raised these points, I want to conclude with that I really like the paper and would recommend publication with only minor revision for the case that the authors would like to revise their manuscript a bit in regard to the above points.

Reviewer #4 (Remarks to the Author):

The authors report a novel and rigorous study on the dependence of glycoprotein distribution on cell membrane curvature, which was induced by either nanostructures or membrane-sculpturing proteins. This is a strong study supported by a large amount of work. In particular, the nanostructures included a variety of geometric shapes and curvature values. Some concerns and suggestions are discussed below.

1. Page 4: “we observed that in areas where some nanopillars were accidentally scratched off (areas circled in dashed yellow lines in Fig. 1D)”

The nanopillars that were scratched off are still visible in the bright field image, suggesting some remaining (e.g., a pillar broken in the middle) or opposite (e.g., a pit formed in the portions where a pillar is scratched off) topography, which might affect the membrane curvature and MUC1 distribution. Is it possible to image a cell on the boundary of the patterned pillars, which covers both pillars and original planar surface (instead of the scratched area)? This would exclude any unknown effect of scratched pillars. For example, in Fig. 1D why does the circled area have a much higher overall MUC1 intensity (and lower Cell Mask intensity) than the pillar area, considering there are large planar empty spaces among the pillars? This seems inconsistent with Fig. 2I, where the MUC1 intensity on pillar area and planar area are comparable, just with different distributions due to the pillar-induced membrane curvatures.

2. Fig. 5C

The averaged normalized intensity is uniform on the nanoX top planar surface (except the dark center area), which is higher than the intensity on nanoX outlines (contours).

This seems inconsistent with Fig. S12 A and Fig. 2E, where the fluorescence intensity is concentrated on the nanoX outlines, the X-shape center and arms are all dark.

Minor concerns

1. Page 3: "Although it is now established that overexpression of bulky glyocalyx proteins can induce membrane protrusions, the reciprocal relationship ..." This study revealed that the glycoprotein prefers negative membrane curvature. Do these findings provide a new angle to explain the mechanism of glycoproteins inducing protrusions? Could you comment on this mutually beneficial relationship between glycoprotein and protrusions? How are they a pair of mutual causes and effects?

2. Page 3: "Previous studies by us and others show that when cells are cultured on substrates with vertical nanopillars, the plasma membrane wraps around nanopillars to create local membrane curvatures (Fig. 1B)." Fig. 1B only shows a schematic that the membrane closely wraps the nanostructures. Are there experimental verifications of this basic assumption? The membrane might either closely or loosely wrap the nanostructures, how is this geometry-dependent? This would also provide the rationale of the geometric design (e.g., aspect ratio and pillar height).

For example, Page 6: "For dense nanopillar arrays, the inter-pillar spacing was small enough to induce the formation of membrane protrusions with negative curvatures in the inter-pillar spaces." How is the

inter-pillar spacing optimized here to ensure that the pillars closely imprint membrane protrusions? It is also possible that the membrane could be almost planar on dense pillar tips if the gap space is too small for the membrane to penetrate.

BTW, is it possible to provide a zoom-in overlay in Fig. S7 to compare with that in Fig. 2I?

3. Page 5: "MUC1_42TR and MUC1_21TR show spatial avoidance of the nanobar ends." From Fig 1M, it is difficult to see the avoidance on nanobar ends. Would it be helpful if providing the nanobar outline as a guide for the eye? With precise positions indicated by the outline, it might show that the intensity is relatively weaker on the two ends than over the whole bar region (just a suggestion).

4. Fig. 1E, I, Fig. 2A show schematics of marked outlines for the ratio calculation. Could you provide the actual regions of interest? For example, in Fig. 1E, is the ROI a circle-shape region, or a ring-shape covering only the pillar outline and excluding the top surface?

For all above nanostructures, the authors considered the membrane curvature induced in the lateral X-Y plane, not the vertical Y-Z plane where a positive curvature also exists. But for Fig. 2H, the dense pillar-induced protrusions were actually in the vertical plane, not lateral X-Y plane. Does the vertical curvature also play a role (contribute to the intensity on nanostructure top surfaces) in the previous pillar/nanobar/nanoX cases where mainly X-Y plane curvatures were considered (intensity mostly concentrated in the outlines)? If there is such a crosstalk between vertical and lateral membrane curvatures, could it be minimized by a good focusing in the confocal microscopy (vertical resolution $\sim 0.5\mu\text{m}$)? i.e., If it is ideally focused on a cross-section of the nanostructure, would the imaging show only the lateral X-Y plane signals, without the effect from vertical curvature? Could you comment on this?

Haogang Cai

RESPONSES TO REVIEWERS' COMMENTS

We thank the reviewers for their constructive and critical comments that have been VERY helpful in improving the quality of our manuscript. In response to reviewers' comments, we performed many new experiments, added new analyses, and substantially revised the main text and the figures. Below are the point-by-point responses to reviewers' comments.

To all reviewers: before addressing individual reviewer's comments, we would like to provide clarification on the data analysis. Reviewers' comments made us realize that we didn't clearly explain the quantification method and how the conclusions were drawn. Here, we describe step-by-step from the raw images to the quantified values, with detailed statistics of the number of nanobars, the number of cells, and the number of experiments.

Figure R1 shows data processing steps for a fixed U2OS cell co-expressing MUC1_42TR-GFP and mCherry-CAAX. **Step 1:** load three-channel (GFP, mCherry, and bright field) images taken from the same field. **Step 2:** Manually click on the center of three nanobars (red arrows) in the mCherry-CAAX channel. Since the distance between nanobars is fixed, the software automatically locates all the nanobars in a rectangular array (yellow circles). Next, the software automatically propagates the nanopillar locations to all three color channels (GFP, mCherry, and bright field). **Step 3:** By intensity thresholding in the GFP channel, the software removes nanobars that are located outside the cell of interest. In some cases, a nanobar outside the cell needs to be manually removed by clicking anywhere inside its yellow circle. **Step 4:** Based on the nanobar locations, the software automatically creates an averaged nanobar image from all the nanobars inside a cell. The cell in the example image interacts with 100 nanobars. In general, each cell contacts ~30-150 nanobars. An average nanobar image is created for each color channel for the selected cell. **Step 5:** Four ROIs, two located at the ends of the nanobar and two located at the side walls of the nanobar, are created on the membrane mCherry channel of the averaged nanobar image. The same ROI locations are re-created on the GFP channel. The same ROIs are used for all cells. **Step 6:** From the ROIs, the nanobar end/side ratios are independently calculated for mCherry and GFP channels. Then, the ratio for the MUC1_42TR-GFP channel (green) is divided by the ratio for the mCherry-CAAX channel (red). This step normalizes the protein ratio to the membrane ratio, which helps to distinguish whether the protein truly has a curvature preference vs. whether there is a higher protein signal due to more membranes at curved locations.

Step 1

Step 2

Step 3

Step 4

Step 5

Step 6

Figure R1: Step-by-step illustration of nanobar data processing and quantification. Three channels- mCherry-CAAX, MUC1-GFP, and bright field, from the same location are needed. This step-by-step quantification process is now provided in Suppl. Figure S2.

Figure R2 shows the example image and the averaged images of nanobars over many cells with detailed statistics shown in **Table R1**. As in the example images of MUC1_42TR and alpha-AP2, some of the nanobars show clear protein accumulation (responsive) while others do not (non-responsive). In order to avoid any user-bias, we included all nanobars inside cells when calculating the average image. Including the non-responsive nanobars likely increases the overall background of the average image and makes the curvature effect less visible to the eye. Quantification of the end/side ratios over many cells shows that the differences between MUC1_42TR/MUC1_21TR and CAAX are statistically significant ($p < 0.0001$). In the revised manuscript, we also include data points for individual cells, with each cell data representing the average of 50-100 nanobars.

Figure R2: Example averaged nanobar images, and the quantification of protein distributions on nanobars.

Target	Normalized nanobar end-to-side ratio - U2OS cells on 200-nm nanobars					Corresponding figure
	Mean	SD	SEM (SD/ \sqrt{N})	N (# cells)	n (# nanobars)	
α -AP2	2.043	0.647	0.138	22	290	Fig. 1J
F-actin	1.384	0.371	0.085	19	157	
CAAX	1.000	0.038	0.003	170	9092	
MUC1 Δ CT-0TR	1.028	0.047	0.008	38	1757	
MUC1 Δ CT-10TR	0.999	0.042	0.006	50	2097	
MUC1 Δ CT-21TR	0.900	0.047	0.007	48	2202	
MUC1 Δ CT-42TR	0.889	0.043	0.005	63	3415	

Table R1: Detailed statistics for quantifications shown in Figure R2D. The substrate that was used to measure AP2 and F-actin has many nanobars scratched off. There were fewer nanobars per cell.

Point-by-point responses to reviewers' comment

Reviewer #1 (Remarks to the Author):

The MS by Lu et al. describes the ability of MUC1 (one of the major components of the mammalian glycocalyx) to sense the curvature of the plasma membrane and prefer/avoid areas of respectively -ve/+ve curvature. MUC1's curvature preference is suggested to reduce its endocytic uptake. Overall the MS is very well written, and referenced, and the figures are very well presented. Mechanistically MUC1-curvature sensing appears to bear strong similarities with IDPs as investigated by Stachowiak et al. They have demonstrated in two NatComm papers first that IDPs help induce curvature and then they can sense it. It appears that the curvature inducing properties of MUC1 have already been documented. One could claim that curvature sensing by MUC1 is rather predictable, and not that exciting at least from a biophysical perspective. However, it would appear that there is sufficient biological interest in demonstrating curvature sensing by MUC1 and by extension glycoproteins, as e.g. in the case of endocytosis. However, as I explain below in detail, I have major concerns on whether many of the claims in this MS are supported by the data, thus I am unfortunately not able to recommend publication.

We strongly agree with the reviewer that MUC1's curvature sensing is likely the same IDPs and dominated by entropic forces. We added discussions of IDPs in the introduction of the similarity. *“Recent studies show that, due to conformational entropy, intrinsically disordered regions (IDRs) are able to sense membrane curvatures when artificially tethered to the membrane and can amplify the curvature sensitivity of BAR domains (Zeno et al. 2018; Snead et al.). Therefore, MUC1 may sense membrane curvatures via a similar mechanism.”*

1. Claim: “On the same nanobars, MUC1_42TR-GFP appears to distribute more on the flat side walls (Fig. 1K)”. Upon close inspection of Fig. 1K and especially of the averages in Fig. 1M, I see uniform distribution for 21TR and 42TR. Which appears to be in contradiction with the analysis shown in Fig. 1N. The authors should comment on this apparent discrepancy.

We thank the reviewer for commenting about the “*apparent discrepancy*”, which makes us realize that we didn't clearly explain the quantification method and how the conclusions were drawn. Reviewer 2 has similar comments about the quantification. We apologize for the poor explanation/presentation in the previous version of the manuscript. As shown in Figures R1 and R2, we describe step-by-step from the raw images to the quantified values, with detailed statistics of the number of nanobars and the number of cells. In Figure R3 below, we show enlarged average images of all the proteins. We can see visible differences between 21TR/42TR and the CAAX membrane control. The quantified end/side ratios shown in Fig. R2 are normalized against CAAX. In the revised manuscript, we also include data points for individual cells in Fig. R2, with each cell data representing the average of 50-100 nanobars. The difference between 21/42TR and CAAX is statistically significant ($p < 0.0001$).

To clarify, we modified the sentence to “On the same nanobars, MUC1_42TR-GFP appears to distribute more on the flat side walls as compared to the CAAX membrane control.”.

Figure R3: Enlarged average images of different proteins on nanobars. CAAX shows a slightly stronger signal at nanobar ends because of increased membrane area at the curved location.

2. Claim: “In contrast, on the same nanoXs, MUC1_42TR-GFP accumulated more at the inner faces with negative curvatures.”. After close inspection of Fig. 2C-D, and of the averages in 2E/S5, I am not able to confirm the claim about 42TR. Actually, in S5 intensities appear higher on the edges. Which appears to be in contradiction with the analysis shown in Fig. 2F. The authors should comment on this apparent discrepancy.

We thank the reviewer for pointing this out. We should have been more precise in our claims. The claim for MUC1_42TR-GFP is relative to the CAAX membrane control. We modified the claims to “**In contrast, on the same nanoXs, MUC1_42TR-GFP accumulated more at the inner faces with negative curvatures as compared to the CAAX membrane control.**”.

The quantification of nanoXs is similar to the quantification of nanobars for step 1-4. After obtaining the average nanoX images for MUC1-GFP and mCherry-CAAX channels for each cell, 16 ROIs are selected on the membrane channel as shown in **Fig. R4**. The same ROI locations are applied to both channels and for all cells. Ratios are separately quantified for the MUC1-GFP channel and the mCherry-CAAX channel. Then, the ratios for proteins (GFP channel) are normalized against the ratios for membranes (mCherry channel). **The normalization against the membrane is important because a protein may appear to accumulate at high positive curvature locations, such as mCherry-CAAX showing higher intensities at the ends of nanoXs (Figure R4), but the effects are due to membrane wrapping and 2D projection instead of a real curvature effect.** If a membrane protein such as MUC1_OTR-GFP does not have a curvature preference, the ratios will be similar to CAAX, and thus the normalized ratios will be close to 1. On the other hand, the ratios for 42TR are

statistically different from CAAX, and the trend is opposite to F-actin. We updated the quantification to include data points for individual cells (Fig. R4).

Figure R4: The data processing for nanoX and the quantified protein distributions on nanoX.

3. In the experiments with I-, F-bars, according to the way the authors have interpreted the results from fig. 1 and 2, one would expect that truncation would produce an antisymmetric response (up/down concentration) to the +/- curvatures. Such an antisymmetric response would validate the mechanism. However, the data show that the colocalization of MUC1 and I-BAR is not at all affected by the length of the ectodomain. Similarly, the triple mutation did not affect the colocalization of MUC1-Ts with IRSp53. To the same degree that the experiments with the F-BAR validate the mechanism, the lack of response in the I-BAR experiments refutes it. This is a significant problem that is neither acknowledged nor addressed, let alone resolved.

This is an important point that was not clearly articulated in the previous version of the manuscript. Here, we clarify how we quantified the colocalization. Our colocalization measurements are automatically processed by image registration of two color channels after background subtraction, followed by pixel-by-pixel correlation using the built-in function of ImageJ, which calculates a Pearson's correlation coefficient (PCC) for each pair of images. Unlike the nanobar end/side ratios that depend on the intensity, the colocalization is binary yes/no for each pixel. For example, MUC1_42TR-GFP strongly prefers negative curvature of

filopodia. MUC1_0TR-GFP does not have a curvature preference and thus distributes evenly on filopodia and flat membranes. Therefore, in IRSp53 co-transfected cells, we expect that MUC1_42TR-GFP and MUC1_0TR-GFP show similar colocalizations with IRSp53 that is mostly on filopodia. On the other hand, MUC1_42TR-GFP is largely absent on FBP17-labeled membrane invaginations, while MUC1_0TR-GFP distributes evenly on membrane invaginations and flat membranes. Therefore, in FBP17 co-transfected cells, MUC1_42TR-GFP and MUC1_0TR-GFP will show different colocalizations with FBP17.

The colocalization between MUC1 and IRSp53 is much less than 100%. This is because, unlike CAAX that shows very little intracellular retention, there is always a fraction of MUC1 proteins trapped in ER or other intracellular organelles, similar to previous reports (Pan et al. 2019; Hisatsune et al. 2009). Nevertheless, there is a significant difference between MUC1 correlations to IRSp53 vs. FBP17 - the Pearson correlation coefficient (PCC) between MUC1_42TR-GFP and IRSp53 is significantly higher than that between MUC1_42TR-GFP and FBP17.

To further address this comment, we quantified a different colocalization parameter, thresholded Mander's overlap coefficient (MOC), which is also calculated in ImageJ similar to Pearson's correlation coefficient (PCC). MOC has been shown to provide similar but sometimes complementary information than PCC (Dunn et al. 2011). Using the same set of data used for PCC calculation, the new MOC quantifications show the same trend as the PCC (**Fig. R5**). PCC is more widely used in literature, thus we keep using PCC quantifications in Fig. 3E, 3F, 4E, 4F and S12C.

Figure R5: Comparisons of PCC (A&B) and MOC (C&D) quantifications for MUC1 localizations with FBP17 and IRSp53. The PCC data is shown in the main Figure 3E&F.

4. Claim: “In sharp contrast, MUC1_42TR-GFP prefers the flat sidewalls of thin nanobars but avoids the two ends with positive curvatures (Fig. 7E).” I do not think that the data I see in 7E or in 7G support this claim. This appears to be in contradiction with the analysis shown in Fig. 7I. The authors should comment on this apparent discrepancy.

We should have been more precise in our claims. We are referring to the protein distribution normalized to the membrane distribution. From Fig. 7E, MUC1_42TR-GFP accumulates to the flat sidewalls of some thin nanobars, while other nanobars do not show clear MUC1_42TR-GFP signals. We averaged all nanobars of the same size to obtain the averaged image of MUC1_42TR shown in Fig. 7G. The averaged image of CAAX from the co-transfection experiment was also obtained in the same way (Fig. S17). The quantified end-to-side ratios of MUC1_42TR shown in Fig. 7I are normalized against the CAAX ratios. To clarify, we revised

the sentence to **“For thin nanobars, in contrast to the membrane marker mCherry-CAAX that shows slightly higher intensity at the ends of nanobars, MUC1_42TR-GFP shows a lower intensity at the ends of nanobars than at the flat sidewalls (Fig. 7E, white arrowheads).”**.

5. Claim: “Through these quantitative measurements, we determined the curvature threshold, or the cutoff value, to be $\sim 1/1200$ nm for MUC1_42TR-GFP’s avoidance of positive curvatures. The threshold diameter at 1200 nm is much larger than the diameter of clathrin-coated pits (50-200 nm).” My understanding of the way entropic forces-based curvature sensing works is based on the data from the group on IDPs. There they do not see, nor expect to see, thresholds but rather 1) monotonic dependencies that 2) get steeper as curvatures increase. Both of these features appear to contradict the findings of Fig. 7 suggesting a different mechanism. The authors should comment on this apparent discrepancy.

We agree with the reviewer that the mechanism should be similar to IDPs via an entropic exclusion. We also agree with the reviewer that “monotonic decrease” is a more accurate description than “threshold”. Even our previous studies that report “threshold” for curvature-sensing proteins are actually monotonic decays instead of on-off switches. We revise the sentence to **“Through these quantitative measurements, we determined the curvature range with an upper diameter boundary ~ 1200 nm for MUC1_42TR-GFP’s avoidance of positive curvatures. The upper diameter boundary at 1200 nm is much larger than the diameter of clathrin-coated pits (50-200 nm).”**

6. Claim: “For this purpose, we designed and fabricated the gradient “NanoX” arrays which are able to induce positive curvatures at the arm ends and negative curvatures at the inner faces (Fig. 2A). ” The authors should provide support for the claim that the membrane conforms to the X shapes.

The plasma membrane adheres tightly to the ends of nanoX, but likely does not conform to inner surfaces of nanoXs. From previous electron microscopy studies (Santoro et al. 2017), we learn that the cell membrane attaches tightly to protruding surfaces, but does not adhere to invaginating surfaces. This can be seen from the membrane marker mCherry-CAAX signals in Figure 2E, which shows a much higher intensity at the four ends than the inner sides. Although negative curvatures will still be induced at the inner faces of nanoXs, the curvature value can not be precisely defined. This is also why there is no apparent difference between the two complementary inner surfaces (e.g. 30° vs. 150°) in nanoXs (Fig. 2F-G). To avoid confusion, we added the following text to the manuscript **“The plasma membrane adheres to the ends of nanoX, but likely not tightly to inner surfaces of nanoXs. This can be seen from mCherry-CAAX signals (Fig. 2E and Suppl. Fig. S6B and S7), which shows a much higher intensity at the four ends than the inner sides. Although negative curvatures are induced at the inner faces of nanoXs, the curvature value is not defined by the angle, which**

explains why there is no apparent difference between the two complementary inner surfaces in nanoXs (Fig. 2F-G).”.

7. Claim: “For dense nanopillar arrays, Fig. 2I, the inter-pillar spacing was small enough to induce the formation of membrane protrusions with negative curvatures in the inter-pillar spaces.” The authors should provide data to support this claim.

We have previously used TEM to investigate how cells interface with nanopillar arrays with the dimension used in this study (both sparse and dense arrays). The image shown in Figure R6 is copied from Figure 3 in (Hanson et al. 2012), which shows a cell sitting on top of dense and large nanopillars without membrane wrapping. The cell used in the TEM study does not have filopodia protrusions. On the other hand, HeLa cells that are shown in Fig. 2I and Fig. S9 (Fig S7 in the old version) have extensive filopodial protrusions. MUC1 preferentially locates on filopodia as previous studies show (Bennett et al. 2001; Hattrup and Gendler 2008). We believe that these protrusions can grow into the inter-pillar space, which is why MUC1 is located at inter-pillar spaces as shown in Fig. 2I. We also provided the zoom-in overlay in Fig. S9, which confirms the same observation.

Figure R6: TEM imaging of cells cultured on nanopillar arrays of different dimensions. The images are copied from Hanson et al, *Nano Letter* (2012).

8. Claim: “Fluorescence recovery after photobleaching (FRAP) measurement shows that supported lipid bilayers formed on SiO₂ nanostructures exhibited similar fluidity as on flat areas

(Suppl. Fig. S12)". S12 does not compare FRAP curves on the two areas and thus does not support this claim.

We thank the reviewer for pointing out the missing control. We now added the result of the FRAP experiments performed on flat surfaces vs. on nanoXs (Figure R7). The FRAP experiments show that the lipid fluidities on both nanoXs and flat surfaces are comparable. This result is consistent with our earlier work and with reports from other research groups (Zhao et al. 2017; Su et al. 2020). The Figure S12 in the previous version becomes Figure S14 in the new version.

Figure R7: FRAP measurements of lipid diffusion on flat and on nanoX areas.

General comments:

The prevailing natural phenotype in the spatial distribution of MUC1, as seen in all cell figures in this MS, appears to be puncta. The authors should comment on this and if possible link them to curvature. Work with IDPs would suggest that surface density, i.e expression levels, strongly influence the coupling to curvature. Have the authors quantified the expression levels of the different mutants and if so what where they are, and where they are the same. It would be very helpful to provide info on the number of different cells and numbers of pillars used on the averages, in addition to the number of independent repeats that should be ≥ 3 in all data sets (especially when SEM is used to reduce s.d.).

We thank the reviewer for requesting details of statistical analysis. We should have provided this in the initial submission. In the revised manuscript, we added a table, Table 1-7 in Supplementary Materials, that includes the number of cells, the number of nanobars/nanoXs/nanopillars, the mean, the standard deviation, and the standard error of the mean, for each quantification value shown from Figure 1 to Figure 7, as well as from supplementary figures.

To address the reviewer's comment about expression levels for different mutants, we performed new flow cytometry experiments. All MUC1 mutants are labeled with a GFP tag. Flow cytometry of 7-different MUC1 mutants show that all MUC1 Δ CT-GFP constructs are expressed at a similar level on cell surfaces (Fig. R8, added as Suppl. Figure S1 in Suppl. Materials). This result agrees well with the previous findings (Shurer et al. 2019).

Figure R8: Flow cytometry measurements of MUC1 expression levels on cell surface of U2OS cells.

For MUC1 immunostaining on HeLa cells, we employed Anti-MUC1/episialin Antibody, clone 214D4 (Sigma-Aldrich) to probe the extracellular domain of the endogenous MUC1. The reviewer is correct that the prevailing natural phenotype of MUC1, probed by 214D4, appears to be puncta. A recent study shows the natural MUC1, as probed by 214D4, is almost exclusively distributed on the membrane protrusions with negative curvatures (Li et al. 2021). We added new texts in the main text to suggest a possible link between natural MUC1's puncta distribution, its relation to curvature, and the implication in MUC1 endocytosis.

Reviewer #2 (Remarks to the Author):

The manuscript titled: "Membrane curvature regulates the spatial distribution of bulky glycoproteins" by Lu et al examines the biophysical behavior of MUC1, a bulky glycoprotein implicated in a range of important biological functions, in curved membranes. The authors used a powerful combination of nanoarray technologies and molecular biology to enforce curvatures on cell membranes with tunable geometries to evaluate how perturbations in membrane morphology affects MUC1 distribution in the membranes.

This work builds on a recent discovery that bulky mucins have the capacity to enforce curvature in membranes and this work shows that this relationship is reciprocal and that membrane curvature may likewise lead to reorganization of the mucinous glycocalyx, which can have biological consequences, such as increased resistance of mucins to enter endosomes leading to their prolonged cell surface residence.

The key takeaways from this study are the findings that bulky mucins avoid membranes with negative curvatures and that this propensity increases with the bulk of the mucin ectodomain (as defined by its length and level of glycosylation) and membrane density (shown in supported lipid bilayers).

The methods are well designed and the data are clearly presented. All appropriate controls are included and the manuscript is very well written and easy to follow.

The paper provides important new insights into the mechanobiology of the mucinous glycocalyx and I recommend it is suitable for publication as is.

We thank the reviewer for the positive comments about this work!

I only had two minor comments for the authors to consider.

1. In certain instances, I found it difficult to visually confirm the changes in Muc1 localization on the nanopillar nanobar arrays using the fluorescence micrographs in Figures 1 and 2. For instance in Fig 1 K and M, I was having a hard time observing the avoidance of the positively curved ends of the nanobar by MUC1-42TR. The distribution of the protein appears quite similar to that of CAAX. Similarly, the distributions of MUC1 in Fig 2 D and E are quite difficult to assess visually (the additional SI heatmaps help). I understand that the authors applied a custom MATLAB program to quantify changes in fluorescence. Perhaps a description of how that process works in the SI would be helpful.

This comment makes us realize that we didn't clearly explain the quantification method. Reviewer 1 has similar comments about the quantification. We apologize for the poor explanation/presentation in the previous version of the manuscript. As shown in Figures R1 and R2, we describe step-by-step from the raw images to the quantified values, with detailed statistics of the number of nanobars and the number of cells. In Figure R9, we show enlarged average images of all the proteins. We can see visible differences between 21TR/42TR and the

CAAX membrane control. The quantified end/side ratios are normalized against CAAX. We also include data points for individual cells in Fig. R9, with each MUC1 cell data representing the average of 40-100 nanobars. The difference between 42TR and CAAX is statistically significant ($p < 0.0001$). The distributions of MUC1 in Fig 2D and 2E are similarly processed. We added the step-by-step quantification process in Suppl. Figure S1.

Figure R9: Averaged nanobar images, and the quantification of protein distributions on nanobars.

2. My second comment relates to the role of the cytosolic domain of Muc1 in the membrane localization process. The authors comment on its role in endocytosis and signaling. I wonder to what extent the interactions of the cytosolic domain with the cytoskeleton may counteract the

avoidance of positive curvatures by MUC1s. Or can the avoidance of such membrane regions drive the reorganization of the cytoskeleton?

To address this comment, we carried out a new set of co-transfection experiments using full-length MUC1_42TR-GFP (with intracellular domain) and IRSp53 or FBP17. The new data shows that full-length MUC1_42TR-GFP preferentially accumulates at IRSp53-induced negative membrane curvatures, but avoids FBP17-induced positive ones. In terms of curvature preference, full-length MUC1_42TR-GFP behaves similarly to MUC1 Δ CT_42TR-GFP (Figure R10, added as Suppl. Figure S12). Phalloidin co-staining shows that actin filaments are present in IRSp53-induced membrane protrusions that also show strong MUC1 accumulation, but not in FBP17-induced invaginations. These new data are added to Suppl. Fig S12.

Figure R10: Full-length MUC1 with intracellular domain, colocalizing with IRSp53 but not with FBP17.

Reviewer #3 (Remarks to the Author):

The only criticism I have is about the presentation and discussion of the entanglement of whether glycoproteins cause membrane curvature (as shown by the works referenced in the second abstract of the introduction) or are guided by membrane curvature, or more likely what the extend of both effects is. I think the best evidence for the "sensing not sculpturing" is given by the author also adding the in vitro experiments, as there a very simple system without any other components and proteins that could play a role in addition to the MUC1 is presented. So in this context, I was feeling a bit confused when after the the second paragraph (with the references to work showing glycoproteins can cause membrane curvature), the next paragraph first discussed generally about the role of membrane curvature for cell physiology and only then comes back to the contrast "curvature sculpture against sensing" again. So I would suggest to move the general introduction to the role of membrane curvature (the passage "Cell membranes are dynamic [...] cell invasion") to another place, maybe directly after the introduction on MUC1 or even as the overall start of the whole introduction, to have the contrasting section ("Although it is now established [...]") directly following on the introduction of glycoproteins sculpturing curvature. This would in my opinion even more clearly present the main point and novelty of the authors' work.

We thank the reviewer for pointing out the entanglement of the logic flow in the text. We totally agree with the reviewer's suggestion and made substantial changes in the text to avoid the back-and-forth discussion of the curvature sensing and curvature generating. The text reads much better than the previous version.

1. One question to the fluorescence images in Figure 1c. Looking at the a-MUC1 / CellMask channels, it looks like there is a kind of "window" in the cell with overall reduced fluorescence intensity, as only in this region the raised CellMask signal on/near the pillars is clearly visible (while the rest looks more homogeneously red). Additionally , the a-MUC1 signal is raised in these areas to a point where the pillars become much more visible (e.g. in the area right to the dash marked box). Have the authors had some idea of what this effect is?

The "window" is the location where the nucleus is located. The nucleus location often has a lower fluorescence background because there is less background fluorescence from the cytosol or from the intracellular membrane. As a result, signals from the bottom cell membrane appear to be more pronounced. Because MUC1 is a transmembrane protein, it also shows a weak signal at nanopillar locations where the membrane areas are larger than surroundings, albeit at a much lower level than protrusions at the peripheral of the cell.

The reviewer asked why "only in this region the raised CellMask signal on/near the pillars is clearly visible (while the rest looks more homogeneously red)". This is related to the properties of the "CellMask" dye. This dye is supposed to label the plasma membrane. However, unlike the protein marker CAAX that exclusively labels the plasma membrane, it is known that the CellMask dye can penetrate the plasma membrane to label intracellular membranes, depending

on the dye incubation time. The red image in Figure 1C (the previous version) shows a high background due to intracellular membrane labeling. The high background masks the nanopillar effect to make the regions look homogeneous. As a comparison, the plasma membrane labeled by CAAX (Fig. S5A and S5B) shows membrane wrapping on all nanopillars. To clarify, we added in the manuscript “*We note that CellMask can also stain intracellular membranes at a low intensity, thus giving an overall higher background than CAAX.*”. In the revised version of the manuscript, we have substantially rearranged Figure 1. The Figure 1C in the previous version becomes Figure 1K in the new version.

2. And (also Figure 1c), for further clarity, I suggest that the authors include a full set (a-AP2, CellMask, Bright field) of images also for the separate experiment in the fourth panel (a-AP2) - can be done in the supporting information of course to not clutter the Figure 1 even more.

We have now added a full set (a-AP2, CellMask, Bright field) of images in Figure R11, also added as Suppl. Figure S4A.

Figure R11: Fluorescence and bright field images of a-AP2, CellMask, and nanopillars.

Having raised these points, I want to conclude that I really like the paper and would recommend publication with only minor revision for the case that the authors would like to revise their manuscript a bit in regard to the above points.

We thank the reviewer for the positive feedback about this work!

Reviewer #4 (Remarks to the Author):

The authors report a novel and rigorous study on the dependence of glycoprotein distribution on cell membrane curvature, which was induced by either nanostructures or membrane-sculpturing proteins. This is a strong study supported by a large amount of work. In particular, the nanostructures included a variety of geometric shapes and curvature values. Some concerns and suggestions are discussed below.

1. Page 4: “we observed that in areas where some nanopillars were accidentally scratched off (areas circled in dashed yellow lines in Fig. 1D)”

The nanopillars that were scratched off are still visible in the bright field image, suggesting some remaining (e.g., a pillar broken in the middle) or opposite (e.g., a pit formed in the portions where a pillar is scratched off) topography, which might affect the membrane curvature and MUC1 distribution. Is it possible to image a cell on the boundary of the patterned pillars, which covers both pillars and original planar surface (instead of the scratched area)? This would exclude any unknown effect of scratched pillars. For example, in Fig. 1D why does the circled area have a much higher overall MUC1 intensity (and lower Cell Mask intensity) than the pillar area, considering there are large planar empty spaces among the pillars? This seems inconsistent with Fig. 2I, where the MUC1 intensity on pillar area and planar area are comparable, just with different distributions due to the pillar-induced membrane curvatures.

To address this concern, we carried out new experiments and imaged cells that grew at the boundary of nanopillar/flat areas. Similar to the nanopillar scratched-off results, we found that MUC1 prefers flat areas rather than nanopillar areas. The old images are replaced by the new images in Figure 1L in the new version.

Figure R12: MUC1 and CellMask staining of a cell at the boundary of nanopillar/flat areas.

We would like to explain the seemingly inconsistency between Figure 1D and Figure 2I. HeLa cells express a high level of MUC1 and these cells also have extensive filopodia protrusions (unlike U2OS cells). Previous studies show that MUC1 preferentially distributes on these filopodia protrusions with negative curvature. **Figure 1D** (in the old version) (**200-nm diameter, 2.5- μm spacing**) and **Figure 2I** (**500 nm diameter, 1 μm spacing**) differ in their nanopillar array dimensions. From Figure 1D, it appears that filopodia (negative curvatures) preferentially

occur on the portion of the cell that is in contact with the flat areas. We speculate that it is because nanopillars create positively membrane curvatures, which makes it unfavorable for filopodia formation that requires the generation of negative membrane curvatures. For Figure 2I, the nanopillars are bigger and densely packed. The cell membrane is likely not wrapping around the nanopillars (See TEM images shown in Figure R6) such that the inter-pillar space can effectively induce or accommodate membrane protrusions. More discussions about the large and dense nanopillars are included later when addressing another comment.

We added in the text *“Interestingly, when cells were at the nanopillar/flat boundary, MUC1 appeared to prefer the flat region rather than the nanopillar region (Fig. 1D). Previous studies show that MUC1 in Hela cells largely localizes on filopodia protrusions. We speculate that nanopillars create positively membrane curvatures, which makes it unfavorable for filopodia formation that requires the generation of negative membrane curvatures.”*.

2. Fig. 5C: The averaged normalized intensity is uniform on the nanoX top planar surface (except the dark center area), which is higher than the intensity on nanoX outlines (contours). This seems inconsistent with Fig. S12 A and Fig. 2E, where the fluorescence intensity is concentrated on the nanoX outlines, the X-shape center and arms are all dark.

For Fig. 5C and Fig. S12A (previous version of the manuscript), both images show supported lipid bilayers on the nanoX structures. When the focal plane is near the bottom of the nanoX such as Fig. S12A, the X structure shows a dark center. When the focal plane is near the middle height of nanoX such as Fig. S12B, S12C and S12D, the X structure does not show a dark center because the top surface of nanoX is slightly inside the imaging volume. We try to focus near the middle height of nanoX, but sometimes the focal plane deviates slightly such as that in Fig. S11A. Fig. 5C shows the average image of many nanoX structures (averaged over >200 nanobars for each geometry). Therefore, X structures in Fig. 5C do not show a dark center.

Fig. 2E reports the averaged cell membrane instead of supported lipid bilayer. The plasma membrane adheres tightly to the ends of nanoX, but does not conform to inner surfaces of nanoXs. This can be seen from the membrane marker mCherry-CAAX signals in Figure 2E, which shows a much higher intensity at the ends of nanoX than that at the ends of nanobars. For Fig. 2E, we were trying to image F-actin, which shows a much clearer signal near the bottom surface. The focal plane is closer to the bottom of nanoX. So, for Fig. 2E, the X structures show a dark center. From the reviewer’s comment, we realize that the images can be confusing without proper explanation. We added the focal plane explanation to the main text.

Minor concerns

1. Page 3: “Although it is now established that overexpression of bulky glycoalyx proteins can induce membrane protrusions, the reciprocal relationship ...” This study revealed that the glycoprotein prefers negative membrane curvature. Do these findings provide a new angle to explain the mechanism of glycoproteins inducing protrusions? Could you comment on this mutually beneficial relationship between glycoprotein and protrusions? How are they a pair of mutual causes and effects?

This study supports an entropic-force mechanism that links glycoprotein with membrane protrusions. MUC1 possesses a large ectodomain (head) and a single transmembrane coil (tail). At high packing densities, the large ectodomains entropically repel each other, which exert a force through its transmembrane tail to bend the cell membrane, thus generating membrane protrusions. Conversely, on membrane curvatures that are pre-existing, MUC1 at high densities would prefer protrusions that can effectively reduce the repulsion among ectodomains, as shown in **Fig. R13**.

Figure R13: Entropic repulsion of MUC1 leads to membrane bending.

2. Page 3: "Previous studies by us and others show that when cells are cultured on substrates with vertical nanopillars, the plasma membrane wraps around nanopillars to create local membrane curvatures (Fig. 1B)." Fig. 1B only shows a schematic that the membrane closely wraps the nanostructures. Are there experimental verifications of this basic assumption? The membrane might either closely or loosely wrap the nanostructures, how is this geometry-dependent? This would also provide the rationale of the geometric design (e.g., aspect ratio and pillar height).

For sparse nanopillar arrays such as 200-nm diameter and 2.5 μm spacing, the plasma membrane conformally wraps around nanopillars. This is verified by our previous studies using transmission electron microscopy (TEM) (Hanson et al. 2012) and focus-ion-beam/scanning-electron microscopy imaging (FIB/SEM) (Santoro et al. 2017). It is also independently verified by other research groups (Beckwith et al. 2019; Gopal et al. 2019). The reviewer is correct that the wrapping is geometry dependent. For larger and densely packed nanopillar arrays, such as 500 nm diameter and 1 μm spacing, the cell tends to sit on top

of nanopillars (Hanson et al. 2012). For complex nanostructures such as nanoX, the membrane does not adhere tightly to the inner surface as previous FIB/SEM study shows that the cell membrane does adhere tightly to concave structures(Santoro et al. 2017). Although negative curvatures will still be induced at the inner faces of nanoXs, the curvature value can not be precisely defined.

We modified the sentence and added the appropriate references: **"Previous electron microscopy studies by us and others(Hanson et al. 2012; Santoro et al. 2017; Gopal et al. 2019) show that when cells are cultured on substrates with nanopillar arrays with these dimensions, the plasma membrane wraps around nanopillars to create local membrane curvatures (Fig. 1B)."**

Figure R14: FIB-SEM imaging of cells cultured on nanopillar arrays with 200-nm diameter and 2.5- μ m spacing. The images are copied from (Santoro et al. 2017).

For example, Page 6: “For dense nanopillar arrays, the inter-pillar spacing was small enough to induce the formation of membrane protrusions with negative curvatures in the inter-pillar

spaces.” How is the inter-pillar spacing optimized here to ensure that the pillars closely imprint membrane protrusions? It is also possible that the membrane could be almost planar on dense pillar tips if the gap space is too small for the membrane to penetrate. BTW, is it possible to provide a zoom-in overlay in Fig. S7 to compare with that in Fig. 2I?

The reviewer is correct that *“It is also possible that the membrane could be almost planar on dense pillar tips if the gap space is too small for the membrane to penetrate.”* We have previously used TEM to investigate how cells interface with nanopillar arrays with the dimension used in this study (both sparse and dense arrays). The image shown below is copied from Figure 3 in (Hanson et al. 2012), which shows a cell sitting on top of nanopillars without membrane wrapping. The neuronal cell used in this early study does not have filopodia protrusions. On the other hand, Hela cells that are shown in Fig. 2I and Fig. S9 have extensive filopodial protrusions. MUC1 preferentially locates on these filopodia (Bennett et al. 2001; Hattrup and Gendler 2008). We believe that these protrusions can grow into the inter-pillar space, which is why MUC1 is located at inter-pillar spaces as shown in Fig. 2I. We also provided the zoom-in overlay in Fig. S9, which confirms the same observation (Figure R15).

To avoid confusion, we revised the text to **“Using transmission microscopy, we have previously shown that cells adhere to the bottom surface and wraps around thin and sparse nanopillars (200 nm diameter and 2.5- μm spacing), but stay at the top of dense and large nanopillar arrays (500 nm diameter and 1- μm spacing, same as the one shown in Fig. 2H). For dense nanopillar arrays, the inter-pillar spacing can accommodate or induce the formation of membrane protrusions with negative curvatures. The localization of MUC1 at inter-pillar spaces further supports that MUC1 avoids positive membrane curvatures induced by nanostructures while preferring negative membrane curvatures.”**

Copied Figure R6: TEM imaging of cells cultured on nanopillar arrays of different dimensions. The images are copied from (Hanson et al. 2012).

Figure R15: Zoom-in overlay of Figure S7. Confocal images of HeLa cells cultured on dense nanopillar arrays with 500-nm diameter and 1- μ m spacing. Zoom-in overlay shows that MUC1 is primarily located at the space between nanopillars.

3. Page 5: “MUC1_42TR and MUC1_21TR show spatial avoidance of the nanobar ends.” From Fig 1M, it is difficult to see the avoidance on nanobar ends. Would it be helpful if providing the nanobar outline as a guide for the eye? With precise positions indicated by the outline, it might show that the intensity is relatively weaker on the two ends than over the whole bar region (just a suggestion).

The images shown in Fig. 1M are images that are averaged over many nanobars in many cells (detailed statistics provided in the new Table 1). We added the detailed steps to obtain these images at the beginning of the document. The end-to-side ratios are obtained for individual cells. Then the ratios of different proteins are normalized against the mean ratio of the membrane control CAAX. The precise ROI positions that are used to calculate the ratios are shown in Figure R1. The same ROI positions are used to calculate each color channel and each cell. We should have been more precise in our description. We revised the sentence to “**Compared to the membrane marker CAAX, MUC1_42TR and MUC1_21TR show reduced signals at the nanobar ends.**”.

Copied Figure R9: Averaged nanobar images, and the quantification of protein distributions on nanobars.

4. Fig. 1E, 1I and 2A show schematics of marked outlines for the ratio calculation. Could you provide the actual regions of interest? For example, in Fig. 1E, is the ROI a circle-shaped region, or a ring-shape covering only the pillar outline and excluding the top surface?

For Fig. 1E of the previous version of the manuscript, the fluorescence signals are calculated as the average within the circle-shaped regions, not only on the rings. To avoid confusion, we have modified the description and the drawing accordingly.

Figure R16: Illustrations of the imaging process and the quantification for the normalized nanopillar-to-surrounding ratio for a protein of interest. Example in R16B shows a large nanopillar with 200-nm diameter. The intensity is calculated as the mean of the area inside the circle.

5. For all above nanostructures, the authors considered the membrane curvature induced in the lateral X-Y plane, not the vertical Y-Z plane where a positive curvature also exists. But for Fig. 2H, the dense pillar-induced protrusions were actually in the vertical plane, not lateral X-Y plane. Does the vertical curvature also play a role (contribute to the intensity on nanostructure top surfaces) in the previous pillar/nanobar/nanoX cases where mainly X-Y plane curvatures were considered (intensity mostly concentrated in the outlines)? If there is such a crosstalk between vertical and lateral membrane curvatures, could it be minimized by a good focusing in the confocal microscopy (vertical resolution $\sim 0.5\mu\text{m}$)? i.e., If it is ideally focused on a cross-section of the nanostructure, would the imaging show only the lateral X-Y plane signals, without the effect from vertical curvature? Could you comment on this?

The reviewer is right that there will be positive curvature in the Z plane, such as when the membrane wraps around the top plane of nanobars. We believe that the vertical curvature also plays a role by contributing to the intensity from nanostructure top surfaces. We usually focus our confocal microscopy to the middle section of nanostructures, but the top plane may still contribute to the background due to poor vertical resolution in the Z axis. On the other hand, fluorescence from the top plane is expected to be relatively uniform along the length of nanobars. Therefore, it mostly contributes to the background. Furthermore, we calculate the protein curvature preference by normalizing the protein distribution to the membrane, which is important in cases where the membrane does not adhere tightly to the substrate.

REVIEWERS' COMMENTS

Reviewer #2 (Remarks to the Author):

The revised manuscript fully addressed my concerns. I appreciate the additional experiments and clarification of the methods for image analysis. Congratulations to the authors on an excellent and exciting paper!

Reviewer #3 (Remarks to the Author):

All my comments with the previous version were sufficiently responded to and I think the manuscript was significantly improved also in regard to the other reviewers concerns. I recommend publication as is now.

Reviewer #4 (Remarks to the Author):

I appreciate the authors' efforts of additional experiments, and including data analysis and experimental details, which clarified and addressed all the concerns. I recommend publication of the revised manuscript as is.